# Explaining a series of models by propagating Shapley values

Hugh Chen [1], Scott M. Lundberg[2] & Su-In Lee [1✉]

Local feature attribution methods are increasingly used to explain complex machine learning models. However, current methods are limited because they are extremely expensive to compute or are not capable of explaining a distributed series of models where each model is owned by a separate institution. The latter is particularly important because it often arises in finance where explanations are mandated. Here, we present Generalized DeepSHAP (G-DeepSHAP), a tractable method to propagate local feature attributions through complex series of models based on a connection to the Shapley value. We evaluate G-DeepSHAP across biological, health, and financial datasets to show that it provides equally salient explanations an order of magnitude faster than existing model-agnostic attribution techniques and demonstrate its use in an important distributed series of models setting.

[1] Paul G. Allen School of Computer Science and Engineering, University of Washington, Seattle, USA. [2] Microsoft Research, Redmond, WA, USA.
✉email: suinlee@cs.washington.edu

With the widespread adoption of machine learning (ML), a series of models (i.e., where the outputs of predictive models are used as inputs to separate predictive models) are increasingly common. Examples include stacked generalization, a widely used technique[1–4] to improve generalization performance by ensembling the predictions of many models (called base-learners) using another model (called a meta-learner)[5], neural network feature extraction, where models are trained on features extracted using neural networks[6,7], typically for structured data[8–10], and *consumer scores*, where predictive models that describe a specific behavior (e.g., credit scores[11]) are used as inputs to downstream predictive models. For example, a bank may use a model to predict customers' loan eligibility on the basis of their bank statements and their credit score, which itself is often a predictive model[12].

Explaining a series of models is crucial for debugging and building trust, even more so because a series of models is inherently harder to explain compared to a single model. One popular paradigm for explaining models are local feature attributions, which explain why a model makes a prediction for a single sample (known as the *explicand*[13]). Existing model-agnostic local feature attribution methods (e.g., IME[14], LIME[15], KernelSHAP[16]) work regardless of the specific model being explained. They can explain a series of models, but suffer from two distinct shortcomings: (1) their sampling-based estimates of feature importance are inherently variable, and (2) they have a high computational cost which may not be tractable for large pipelines. Alternatively, *model-specific* local feature attribution methods (i.e., attribution methods that work for specific types of models) are often much faster than model-agnostic approaches, but generally cannot be used to explain a series of models. Examples include those for (1) deep models (e.g., DeepLIFT[17], Integrated Gradients[18]) and (2) tree models (e.g., Gain/Gini Importance[19], TreeSHAP[20]).

In this paper, we present Generalized DeepSHAP (G-Deep-SHAP)—a local feature attribution method that is faster than model-agnostic methods and can explain complex series of models that preexisting model-specific methods cannot. G-DeepSHAP is based on connections to the Shapley value, a concept from game theory that satisfies many desirable axioms. We make several important contributions:

- We propose a theoretical framework (Methods section Connecting DeepLIFT's rules to the Shapley values) that connects the rules introduced in ref. [17] to the Shapley value with an interventional conditional expectation set function (with a flat causal graph; i.e., a causal graph where arrows are only drawn between input variables and the output) (ICE Shapley value) (Methods section The Shapley value).
- We show that the ICE Shapley value decomposes into an average over single baseline attributions (Methods section Interventional Shapley values baseline distribution), where a single baseline attribution explains the model for a single sample (explicand) by comparing it to a single sample (baseline).
- We propose a generalized rescale rule to explain a complex series of models by propagating attributions while enforcing efficiency at each layer (Fig. 1b, Methods section A generalized rescale rule to explain a series of models). This framework extends DeepSHAP to explain any series of models composed of linear, deep, and tree models.
- We propose a group rescale rule to propagate local feature attributions to groups of features (Methods section Explaining groups of input features). We show that these group attributions better explain models with many features.

Many feature attribution methods must define the absence of a feature, often by masking features according to a single baseline

sample (single baseline attribution)[13,17,18]. In contrast, we show that under certain assumptions, the correct approach is to use many baseline samples instead (Supplementary Method Section 1.5.3). Qualitatively, we show that using many baselines avoids bias that can be introduced by single baseline attributions (section Baseline distributions avoid bias). Additionally, we show that the choice of baseline samples is a useful parameter which changes the question answered by the attributions (Fig. 1c, section Natural scientific questions with baseline distributions).

We qualitatively and quantitatively evaluate G-DeepSHAP in real-world datasets including biological, health, image, and financial datasets. In the biological datasets[21–24], we qualitatively assess group feature attributions based on gene sets identified in prior literature (section Group attributions identify meaningful gene sets). In the health, image, and financial datasets[25–27], we quantitively show that G-DeepSHAP provides useful explanations and is drastically faster than model-agnostic approaches using an ablation test, where we hide features according to their attribution values (sections Loss attributions provide insights to model behavior, Explaining deep image feature extractors, and Explaining distributed proprietary models). We compare extremely popular model-agnostic methods including KernelSHAP and IME which are unbiased stochastic estimators for the Shapley value[14,16,28] (Supplementary Methods Section 1.5.9).

In practice, G-DeepSHAP can use feature attributions to ask many important scientific questions by explaining different parts of the series of models (Fig. 1d). When features used by upstream models are semantically meaningless (deep feature extraction) or hard to understand (stacked generalization), G-DeepSHAP provides explanations in terms of the original features which can often be more intuitive, especially for non-technical consumers. In addition, G-DeepSHAP enables attributions with respect to different aspects of model behavior such as predicted risk or even errors the model makes (loss explanation). Finally, using the group rescale rule enables users to reduce the dimensionality of highly correlated features which makes them easier to understand (group explanation).

In addition, G-DeepSHAP is the only approach we are aware of that enables explanations of a distributed series of models (where each model belongs to a separate institution). Model-agnostic approaches do not work because they need access to every model in the series, but institutions cannot share models because they are proprietary. One extremely prevalent example of distributed models are consumer scores which exist for nearly every American consumer[11] (section Explaining distributed proprietary models). In this setting, transparency is a critical issue, because opaque scores can hide discrimination or unfair practices.

A preliminary version of this manuscript appeared at a workshop, entitled "Explaining Models by Propagating Shapley Values of Local Components"[29].

In this paper, we improve upon two previous approaches (DeepLIFT[17], DeepSHAP[16]) that propagate attributions while maintaining efficiency with respect to a single baseline. We make two improvements: (1) we compare to a distribution of baselines, which decreases the reliance of the attributions on any single baseline (section Baseline distributions avoid bias) and (2) we generalize the rescale rule so that it applies to a series of mixed model types, rather than only layers in a deep model.

More precisely, a closely related method named DeepSHAP was designed to explain deep models ($f : \mathbb{R}^m \to \mathbb{R}$)[16], by performing DeepLIFT[17] using the average as a baseline[16] (Methods section Differences to previous approaches). However, using a single average baseline is not the correct approach to explain nonlinear models based on connections to Shapley values with an interventional conditional expectation set function and a flat causal graph (i.e., a causal graph where arrows are only drawn between input variables and the output)[30]. Instead, we show that

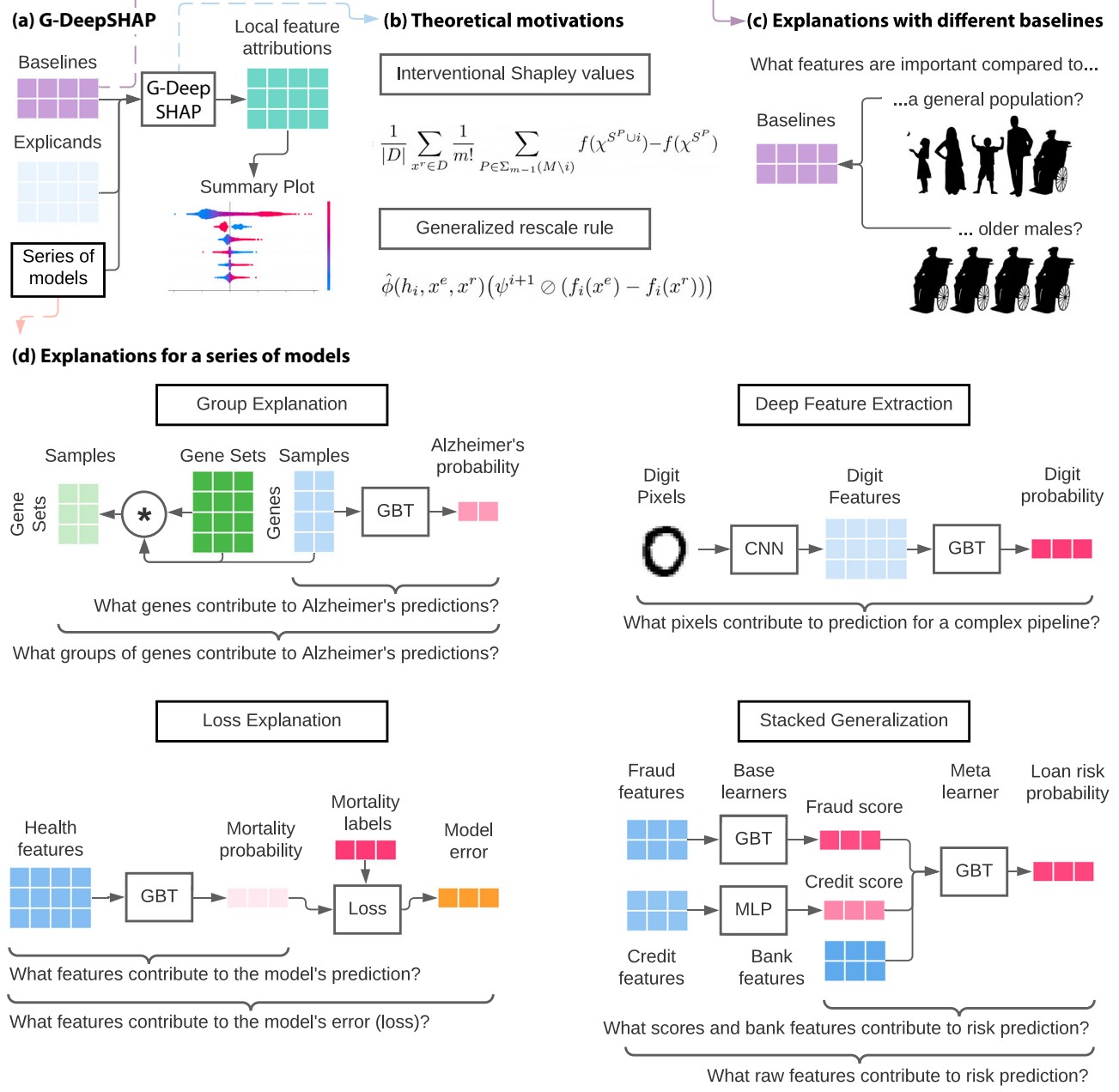

**Fig. 1 G-DeepSHAP estimates Shapley value feature attributions to explain a series of models using a baseline distribution. a** Local feature attributions with G-DeepSHAP require explicands (samples being explained), a baseline distribution (samples being compared to), and a model that is comprised of a series of models. They can be visualized to understand model behavior (Supplementary Methods Section 1.3). **b** Theoretical motivation behind G-DeepSHAP (Methods sections The Shapley value and A generalized rescale rule to explain a series of models). **c** The baseline distribution is an important, but often overlooked, a parameter that changes the scientific question implicit in the local feature attributions we obtain. **d** Explaining a series of models enables us to explain groups of features, model loss, and complex pipelines of models (deep feature extraction and stacked generalization). Experimental setups are described in Supplementary Methods Section 1.2.

the correct way to obtain the interventional Shapley value local feature attributions (denoted as $\phi(f, x^e) \in \mathbb{R}^m$) based on an explicand ($x^e \in \mathbb{R}^m$), or sample being explained, is to average over single baseline feature attributions (denoted as $\phi(f, x^e, x^b) \in \mathbb{R}^m$) where baselines are $x^b \in \mathbb{R}^m$ and $D$ is the set of all baselines (details in Methods section Interventional Shapley values baseline distribution):

$$\phi(f, x^e) = \frac{1}{|D|} \sum_{x^b \in D} \phi(f, x^e, x^b) \quad (1)$$

DeepLIFT[17] explains deep models by propagating feature attributions at each layer of the deep model. Here, we extend DeepLIFT by generalizing DeepLIFT's rescale rule to accommodate more than neural network layers while guaranteeing layer-wise efficiency (details in Methods section A generalized rescale rule to explain a series of models). For a series of models which can be represented as a composition of functions ($f_k(x) = (h_k \circ \cdots \circ h_1)(x)$, where $h_i : \mathbb{R}^{m_i} \to \mathbb{R}^{o_i}$, $m_i = o_{i-1} \forall i \in 2, \cdots k$, $m_1 = m$, and $o_k = 1$) with intermediary models ($f_i(x) = (h_i \circ \cdots \circ h_1)(x)$). In words, $m_i$ are the input dimensions and $o_i$ are the output dimensions for each layer $i$.

G-DeepSHAP attributions are computed as:

$$\psi^k = \hat{\phi}(h_k, x^e, x^b) \quad (2)$$

$$\psi^i = \hat{\phi}(h_i, x^e, x^b)(\psi^{i+1} \oslash (f_i(x^e) - f_i(x^b))), \ i \in 1, \cdots, k-1. \quad (3)$$

We use Hadamard division to denote an element-wise division of $\vec{a}$ by $\vec{b}$ that accommodates zero division, where, if the denominator $b_i$ is 0, we set $a_i/b_i$ to 0. The attributions $\hat{\phi}$ for a particular model in the stack are computed utilizing DeepLIFT with the rescale rule for deep models[17], interventional TreeSHAP for tree models[20], or exactly for linear models. Each intermediate attribution $\psi^i$ serves as feature attribution that satisfies efficiency for $h_i$'s input features, where the attribution in the raw feature space is given by $\psi^1$. This approach takes inspiration from the chain rule applied specifically to deep networks in[17], that we extend to more general classes of models.

G-DeepSHAP is an approximate method, meaning that it is biased for the true interventional Shapley values (Supplementary Notes Section 2.11). However, this bias allows G-DeepSHAP to be drastically faster than alternative approaches. This strategy of trading bias for speed is taken by other Shapley value estimators including L-Shapley[31], C-Shapley[31], Deep Approximate Shapley Propagation[32], and Shapley Explanation Networks[33] (Supplementary Methods Section 1.5). To ensure the attributions are valuable despite this bias, we extensively evaluate G-DeepSHAP both qualitatively and quantitatively in the following sections.

## Results

**Baseline distributions avoid bias**. We now use G-DeepSHAP to explain deep models with different choices of baseline distributions to empirically evaluate our theoretical connections to interventional conditional expectations. We show that single baseline attributions are biased in a CNN that achieves 75.56% test accuracy (hyperparameters in Supplementary Methods Section 1.2.1) in the CIFAR10 data set[34]. We aim to demonstrate that single baselines can lead to bias in explanations by comparing attributions using either a single baseline (an all-black image) as in DeepLIFT or a random set of 1000 baselines (random training images) as in G-DeepSHAP. Although the black pixels in the image are qualitatively important, using a single baseline leads to biased attributions with little attribution mass for black pixels (Fig. 2). In comparison, averaging over multiple baselines leads to qualitatively more sensible attributions. Quantitatively, we show that despite the prevalence of darker pixels (pixel distribution plots in Fig. 2), single baseline attributions are biased to give them low attribution, whereas averaging over many baselines more sensibly assigns a large amount of credit to dark pixels (attribution distribution plots in Fig. 2). To generalize this finding beyond G-DeepSHAP, we replicate this bias for IME and IG, two popular feature attribution methods that similarly rely on baseline distributions (Supplementary Notes Section 2.1).

**Natural scientific questions with baseline distributions**. To demonstrate the importance of baseline distributions as a parameter, we explain an MLP (hyperparameters in Supplementary Methods Section 1.2.2) with 0.872 ROC AUC for predicting 15-year mortality in the NHANES I data set. We use G-DeepSHAP to explain an explicand relative to a baseline distribution drawn uniformly from all samples (Fig. 3a (top)). This explanation places substantial emphasis on age and gender because it compares the explicand to a population that includes many younger/female individuals. However, in practice epidemiologists are unlikely to compare a 74-year-old male to the general population. Therefore, we can manually select a baseline distribution of older

males to reveal novel insights, as in Fig. 3a (bottom). The impact of gender is gone because we compare only to males, and the impact of age is lower because we compare only to older individuals. Furthermore, the impact of physical activity is much higher possibly because physical activity increases active life expectancy, particularly in older populations[35]. This example illustrates that the baseline distribution is an important parameter for feature attributions.

To provide a more principled approach to choosing the baseline distribution parameter, we propose k-means clustering to select a baseline distribution (detail in Methods section Selecting a baseline distribution). Previous work analyzed clustering in the attribution space or contrasting to negatively/positively labeled samples[36]. In Fig. 3b, we show clusters according to age and gender. Then, we explain many older male explicands using either a general population or an older male population baseline distribution (Fig. 3c). When we compare to the older male baselines, the importance of age is centered around zero, gender is no longer important, and the importance orderings of remaining features change. Further, the inquiry we make changes from "What features are important for older males relative to a general population?" to "What features are important for older males relative to other older males?". To quantitatively evaluate whether our attributions answer the second inquiry, we can ablate features in order of their positive/negative importance by masking with the mean of the older male baseline distribution (Fig. 3d, (Methods section Ablation tests)). In both plots, lower curves indicate attributions that better estimated positive and negative importance. For both tests, attributions with a baseline distribution chosen by k-means clustering substantially outperforms a baseline distribution drawn from the general population.

We find that our clustering-based approach to selecting a baseline distribution has a number of advantages. Our recommendation is to choose baseline distributions by clustering according to non-modifiable, yet meaningful, features like age and gender. This yields explanations that answer questions relative to inherently interpretable subpopulations (e.g., older males). The first advantage is that choosing baseline distributions in this way decreases variance in the features that determined the clusters and subsequently reduces their importance to the model. This is desirable for age and gender because individuals typically cannot modify their age or gender in order to reduce their mortality risk. Second, this approach could potentially reduce model evaluation on off-manifold samples when computing Shapley values[37,38] by considering only baselines within a reasonable subpopulation. The final advantage is that the flexibility of choosing a baseline distribution allows feature attributions to answer natural contrastive scientific questions[36] that improve model comprehensibility, as in Fig. 3c.

DeepSHAP (and DeepLIFT) have been shown to be very fast and performant explanation methods for explaining deep models[17,39,40]. In the following sections, we instead focus on evaluating our extension of DeepSHAP (G-DeepSHAP) to accommodate a series of mixed models (trees, neural networks, and linear models) and address four impactful applications.

**Group attributions identify meaningful gene sets**. We explain two MLPs trained to predict Alzheimer's disease status and breast cancer tumor stage from gene expression data with test ROC AUC of 0.959 and 0.932, respectively. We aim to demonstrate that our approach to propagating attributions to groups contributes to model interpretability by validating our discoveries with scientific literature. Gene expression data is often extremely high dimensional; as such, solutions such as gene set enrichment analysis (GSEA) are widely used[41]. In contrast, we aim to

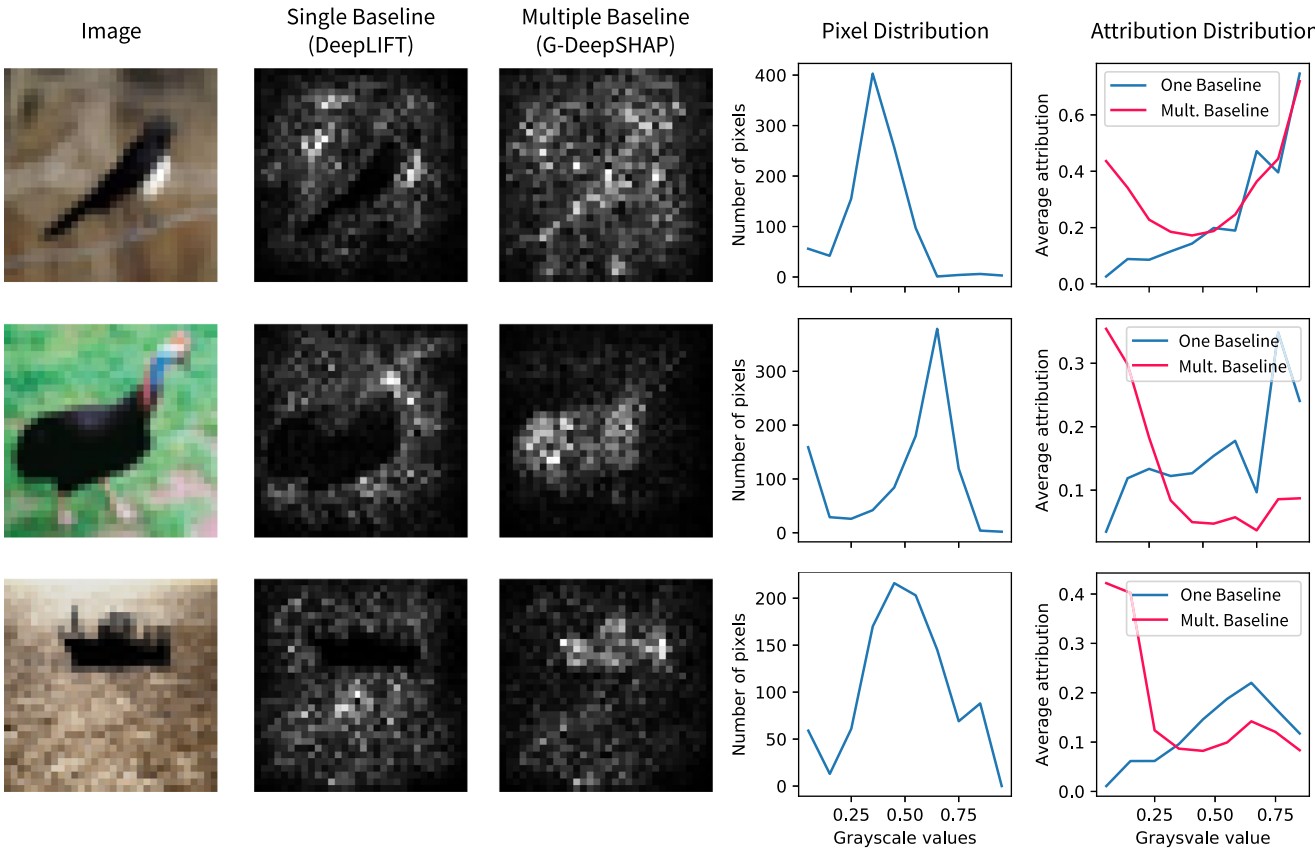

**Fig. 2 Using a single all-black baseline image (DeepLIFT) leads to biased attributions compared to attributions with a randomly sampled baseline distribution (G-DeepSHAP).** The image is the explicand. The attribution plots are the sum of the absolute value of the feature attributions for the three channels of the input image. The pixel distribution is the distribution of pixels in terms of their grayscale values. The attribution distribution is the amount of attribution mass upon a group of pixels binned by their grayscale values.

attribute importance to gene sets while maintaining efficiency by proposing a *group rescale rule* (Methods section Explaining groups of input features). This rule sums attributions for genes belonging to each group and then normalizes according to excess attribution mass due to multiple groups containing the same gene. It generalizes to arbitrary groups of features beyond gene sets, such as categories of epidemiological features (e.g., laboratory measurements, demographic measurements, etc.).

In Fig. 4, we can validate several key genes identified by G-DeepSHAP. For Alzheimer's disease, the overexpression of SERPINA3 has been closely tied to prion diseases[42], and UBTD2 has been connected to frontotemporal dementia—a neurodegenerative disorder[43]. For breast cancer tumor stage, UBE2C was positively correlated with tumor size and histological grade[44]. In addition to understanding gene importance, understanding higher-level importance can be obtained using gene sets, i.e., groups of genes defined by biological pathways or co-expression. We obtain gene set attributions by grouping genes according to curated gene sets from the KEGG (Kyoto Encyclopedia of Genes and Genomes) pathway database (https://www.gsea-msigdb.org/gsea/msigdb/collections.jsp#C2) (additional gene set attributions in Supplementary Notes Section 2.5)

Next, we verify important gene sets identified by G-DeepSHAP. For Alzheimer's disease, the glyoxylate and dicarboxylate metabolism pathway was independently identified based on metabolic biomarkers[45]; several studies have demonstrated aberrations in the TCA cycle in Alzheimer's disease brain[46]; and alterations of purine-related metabolites are known to occur in early stages of Alzheimer's disease[47]. For breast cancer, many relevant proteins are involved in ubiquitin-

proteasome pathways[48] and purine metabolism was identified as a major metabolic pathway differentiating a highly metastatic breast cancer cell line from a slightly metastatic one[49]. Identifying these phenotypically relevant biological pathways demonstrates that our group rescale rule identifies important pathways.

**Loss attributions provide insights to model behavior**. We examine an NHANES (1999–2014) mortality prediction GBT model (0.868 test set ROC AUC) to show how explaining the model's loss (loss explanations) provides important insights different from insights revealed by explaining the model's output (output explanations). G-DeepSHAP lets us explain transformations of the model's output. For instance, we can explain a binary classification model in terms of its log-odds predictions, its probability predictions (often easier for non-technical collaborators to understand; see Supplementary Notes Section 2.4), or its loss computed based on the prediction. Here, we focus on local feature attributions that explain per-sample loss.

We train our model on the first five release cycles of the NHANES data (1999–2008) and evaluate it on a test set of the last three release cycles (2009–2014) (Fig. 5a). As a motivating example, we simulate a covariate shift in the weight variable by re-coding it to be measured in pounds, rather than kilograms, in release cycles 7 and 8 (Fig. 5b). Then, we ask, "Can we identify the impact of the covariate shift with feature attributions?" Comparing the train and test output attributions, release cycles 7 and 8 are skewed, but they mimic the same general shape of the training set attributions. If we did not color by release cycles, it might be difficult to identify the covariate shift. In contrast, for loss

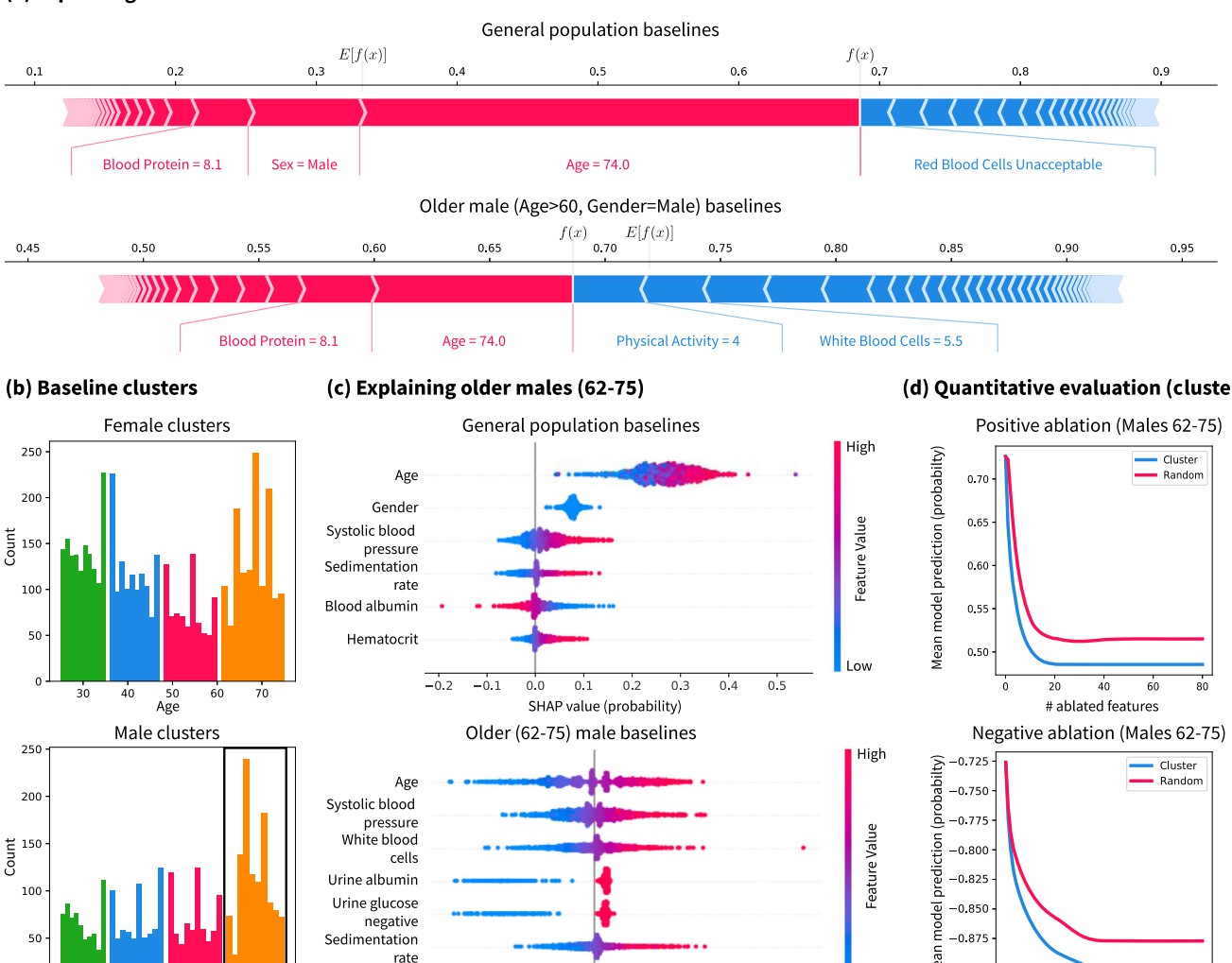

**Fig. 3 The baseline distribution is an important parameter for model explanation. a** Explaining an older male explicand with both a general population baseline distribution and an older male baseline distribution. Red colors denote positive attributions and blue denotes negative attributions. **b** Automatically finding baseline distributions using 8-means clustering on age and gender. Each cluster is shown in a different color. **c** Explaining the older male subpopulation (62–75 years old) with either a general population baseline or an older male baseline. **d** Quantitative evaluation of the feature attributions via positive and negative ablation tests where we mask with the mean of the older male subpopulation (the negative ablation test reports negative mean model output so that lower is better). Note that **b** shows summary plots (Supplementary Methods Section 1.3.3) and **c** shows dependence plots (Supplementary Methods Section 1.3.2).

attributions with positive labels, we can identify that the falsely increased weight leads to many misclassified samples where the loss weight attribution exceeds the expected loss. Although such debugging is powerful, it is not perfect. Note that in the negatively labeled samples, we cannot clearly identify the covariate shift because higher weights are protective and lead to more confident negative mortality prediction.

Next, we examine the natural generalization gap induced by covariate shift over time, which shows a dramatically different loss in the train and test sets (Fig. 5c). We can see that output attributions are similarly shaped between the train and test distributions; however, the loss attributions in the test set are much higher than in the training set. We can quantitatively verify that negative blood lead affects model performance more in the test set by ablating blood lead for the top 10 samples in the train and test sets according to their loss distributions. From this, we can see that blood lead constitutes a substantial covariate shift in the model's loss and helps explain the observed generalization gap.

As an extension of the quantitative evaluation in Fig. 5c, we can visualize the impact on the model's loss of ablating by output attributions compared to ablating by loss attributions (Fig. 5d). This ablation test (Methods section Ablation tests) asks "What features are important to the model's performance (loss)?" Ablating the positive and negative attributions both increase the mean model loss by hiding features central to making predictions. However, ablating by the negative loss attribution directly increases the loss far more drastically than ablating by the output. More so, ablating positive loss attributions clearly decreases the mean loss, which is not achievable by output attribution ablation. Finally, we compare loss attributions computed using either a model-agnostic approach or G-DeepSHAP. In this setting, G-DeepSHAP is two orders of magnitude faster than model-agnostic approaches (IME, Kernel-SHAP, and LIME) while showing extremely competitive positive loss ablation performance and the best negative loss ablation performance.

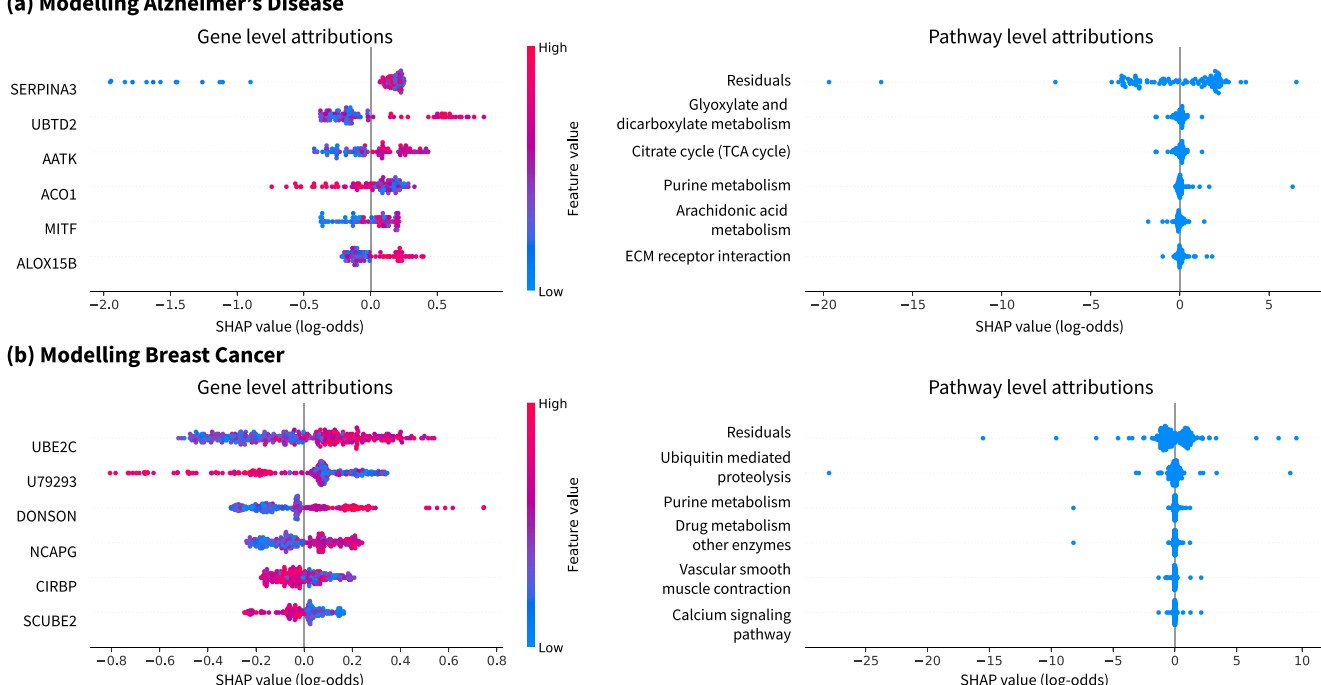

**Fig. 4 Propagating attributions to gene sets enables higher-level understanding. a** Gene and gene set attributions for predicting Alzheimer's disease using gene expression data. **b** Gene and gene set attributions for predicting breast cancer tumor stage using gene expression data. Residuals in the gene set attributions summarize contributions for genes that are not present in any gene set and describes variations in output not described by the pathways we analyzed. Note that (**a**) and (**b**) show summary plots (Supplementary Methods Section 1.3.3).

**Explaining deep image feature extractors**. We compare G-DeepSHAP explanations to a number of model-agnostic explanations for a series of two models: a CNN feature extractor fed into a GBT model that classifies MNIST zeros with 0.998 test accuracy. In this example, nonlinear transformations of the original feature space improve the performance of the downstream model (Supplementary Notes Section 2.6) but make model-specific attributions impossible. Qualitatively, we can see that G-DeepSHAP and IME are similar, whereas KernelSHAP is similar for certain explicands but not others (Fig. 6a). Finally, LIME's attributions show the shape of the original digit, but there is a consistent attribution mass around the surrounding parts of the digit. Qualitatively, we observe that the G-DeepSHAP attributions are sensible. The pixels that constitute the zero digit and the absence of pixels in the center of the zero are important for a positive zero classification.

In terms of quantitative evaluations, we report the runtime and performance of the different approaches in Fig. 6b. We see that G-DeepSHAP is an order of magnitude faster than model-agnostic approaches, with KernelSHAP being the second-fastest. Then, we ablate the top 10% of important positive or negative pixels to see how the model's prediction changes. If we ablate positive pixels, we would expect the model's predictions to drop, and vice versa for negative pixels; doing both showed that G-DeepSHAP outperforms KernelSHAP and LIME, and performs comparably to IME at a greatly reduced computational cost.

**Explaining distributed proprietary models**. We evaluate G-DeepSHAP explanations for a consumer scoring example that feeds a simulated GBT fraud score model and a simulated MLP credit score model into a GBT bank model, which classifies good risk performance (0.681 test ROC AUC) (Fig. 7). Consumer scores (e.g., credit scores, fraud scores, health risk scores, etc.) describe individual behavior with predictive models[11]. A vast

industry of data brokers generates consumer scores based on a plethora of consumer data. For instance, a single data broker in a 2014 FTC study had 3000 data segments on nearly every consumer in the United States, and another broker added three billion new records to its databases each month[50]. As an example of this, the HELOC data set had an ExternalRiskEstimate feature that we removed because it was opaque. Unfortunately, explaining the models that use consumer scores can obscure important features. For instance, explaining the bank model in Fig. 7a will tell us that fraud and credit scores are important (in Fig. 7c), but these scores are inherently opaque to consumers[11]. The truly important features may instead be those that these scores use. A better solution might be model-agnostic methods that explain the entire pipeline at once. However, the model-agnostic approaches require access to all models. In Fig. 7a, a single institution would have to obtain access to fraud, credit, and bank models to use the standard model-agnostic approaches (Fig. 7b (left)). This may be fundamentally impractical because each of these models is proprietary. This opacity is concerning given the growing desire for transparency in artificial intelligence[11,50,51].

G-DeepSHAP naturally addresses this obstacle by enabling attributions to the original features without forcing companies to share their proprietary models if each institution in the pipeline agrees to work together and has a consistent set of baselines. Furthermore, G-DeepSHAP can combine any other efficiency-satisfying feature attribution method in an analogous way (e.g., integrated/expected gradients[18]). Altogether, G-DeepSHAP constitutes an effective way to glue together explanations across distributed models in the industry. In particular, in Fig. 7a, the lending institution can explain its bank model in terms of bank features and fraud and credit scores. The bank then sends fraud and credit score attributions to their respective companies, who can use them to generate G-DeepSHAP attributions to the original fraud and credit features. The fraud and credit institutions then send the attributions back to the bank, which

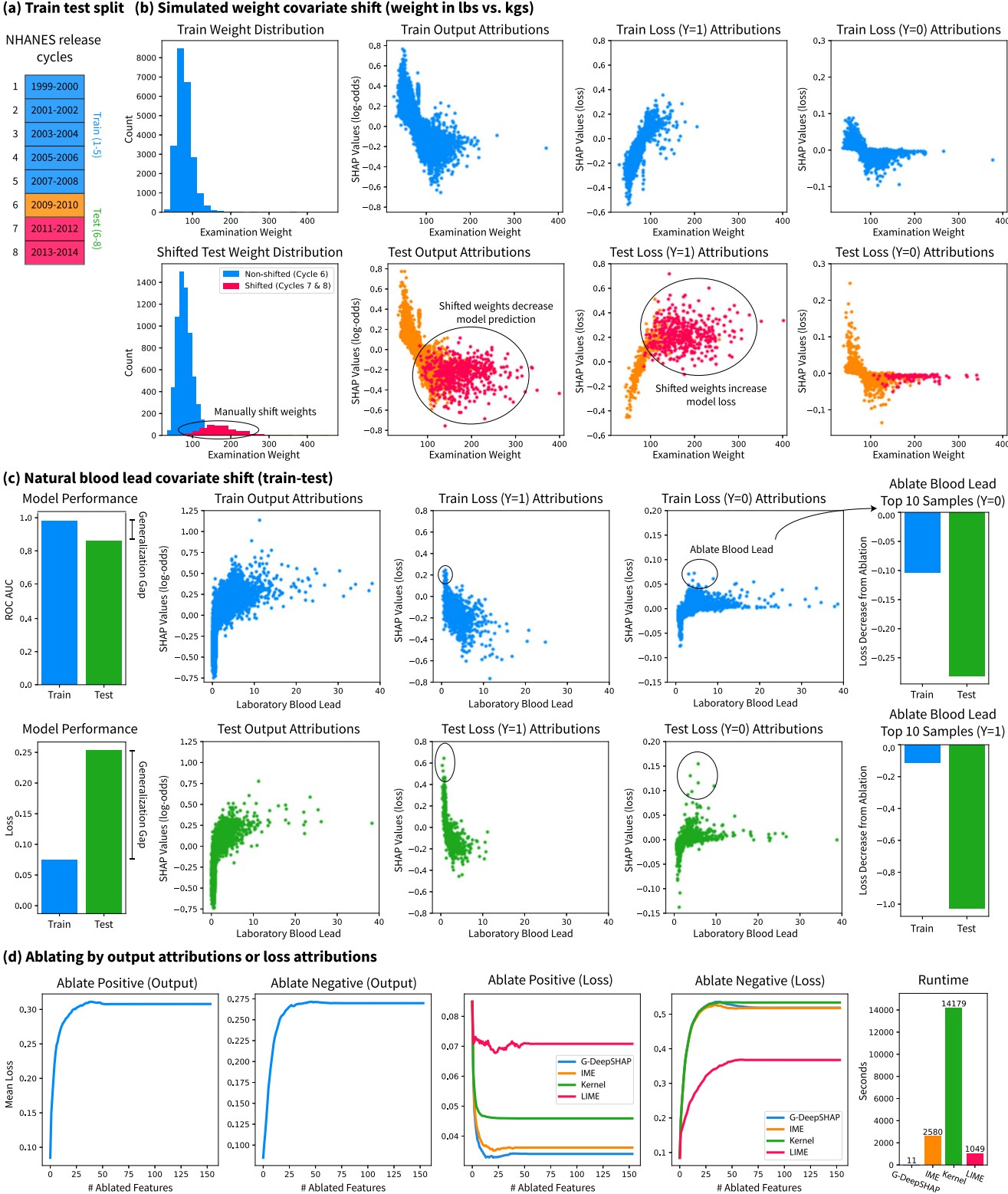

**Fig. 5 Explanations of the model's loss rather than the model's prediction yields new insights. a** We train on the first five cycles of NHANES (1999–2008) and test on the last three cycles (2009–2014). (**b**) We identify a simulated covariate shift in cycles 7–8 (2011-2014) by examining loss attributions. **c** Under a natural covariate shift, we identify and quantitatively validate test samples for which blood lead greatly increases the loss in comparison to training samples. **d** We ablate output attributions (G-DeepSHAP) and loss attributions (G-DeepSHAP, IME, KernelSHAP, and LIME) to show their respective impacts on model loss. We compare only to model-agnostic methods for loss attributions because explaining model loss requires explaining a series of models. Note that (**b**) and (**c**) show dependence plots (Supplementary Methods Section 1.3.2)).

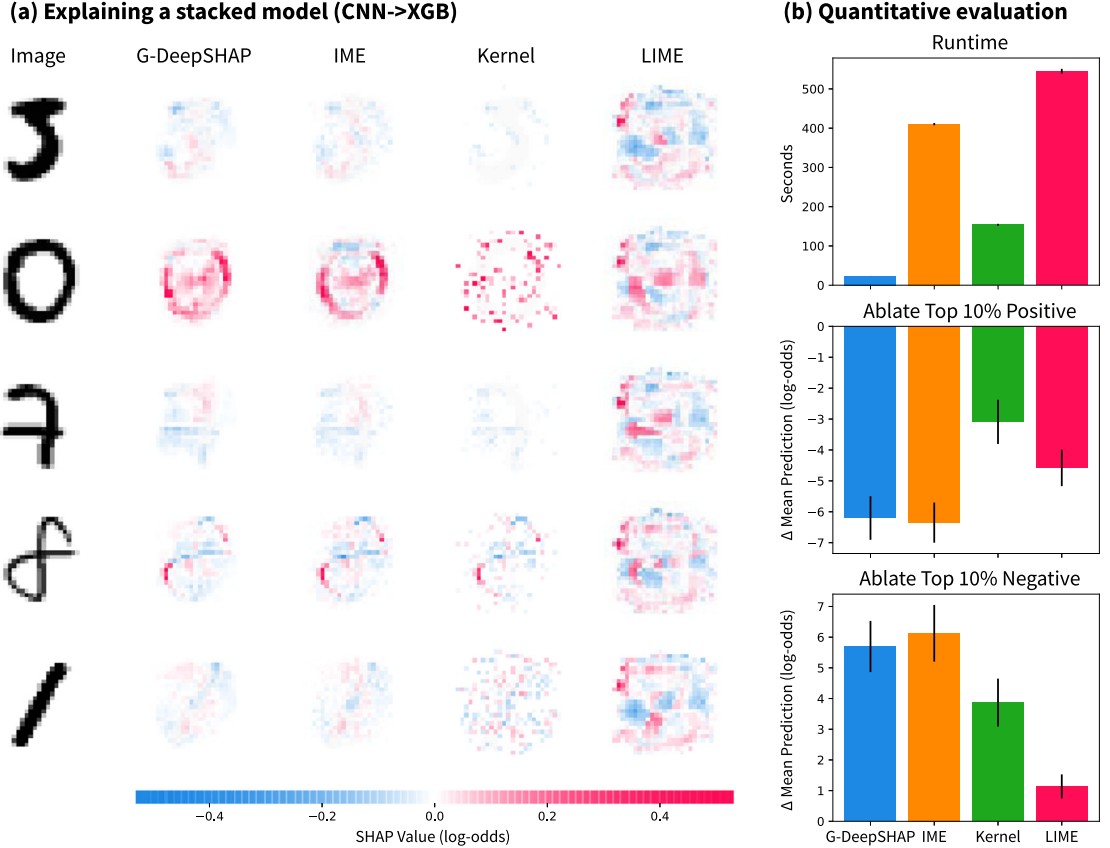

**(a) Explaining a stacked model (CNN->XGB)**

**(b) Quantitative evaluation**

**Fig. 6 Explaining a series of models comprised of a convolutional neural network feature extractor and a gradient boosted tree classifier.** **a** Explanations from G-DeepSHAP and state-of-the-art model-agnostic approaches. **b** Quantitative evaluation of approaches, including runtime and ablation of the top 10% of positive and negative features. Error bars are 95% confidence intervals based on 20 iterations of randomly drawing five explican images, then computing attributions and ablation results.

can provide explanations in terms of the original, more interpretable features to their applicants (Fig. 7d).

We first quantitatively verify that the G-DeepSHAP attributions for this pipeline are comparable to the model-agnostic approaches in Fig. 7b. We once again see that G-DeepSHAP attributions are competitive with the best performing attributions methods for ablating the top five most important positive or negative features. Furthermore, we see that G-DeepSHAP is several orders of magnitude faster than the best performing ablation methods (KernelSHAP and IME) and an order of magnitude faster and much more performant than LIME.

We can qualitatively verify the attributions in Fig. 7c, d. In Fig. 7c, we find that the fraud and credit scores are extremely important to the final prediction. In addition, bank features include low revolving balance divided by credit limit (NetFractionRevolvingBurden) and a low number of months since inquisitions (MSinceMostRecentInqExcl7Days) are congruously important to good risk performance. Then, in Fig. 7d we use the generalized rescale rule to obtain attributions in the original feature space. Doing so uncovers important variables hidden by the fraud and credit scores. In particular, we see that the fraud score heavily relied on a high number of months since the applicants' oldest trade (MSinceOldestTradeOpen), and the credit score relied on a low number of months since recent delinquency (MSinceMostRecentDelq) in order to identify applicants that likely had good risk performance. Importantly, the pipeline we analyze in Fig. 7a also constitutes a stacked generalization ensemble, which we analyze more generally in Supplementary Notes Section 2.7.

## Discussion

In this manuscript, we presented examples where explaining a series of models is critical. Series of models are prevalent in a variety of applications (health, finance, environmental science, etc.), where understanding model behavior contributes important insights. Furthermore, having a fast approach to explain these complex pipelines may be a major desiderata for a diagnostic tool to debug ML models.

The practical applications we focus on in this paper include gene set attribution, where the number of features far surpasses the number of samples. In this case, we provide a rule that aggregates group attributions to higher-level groups of features while maintaining efficiency. Second, we demonstrate the utility of explaining transformations of a model's default output (Supplementary Notes Section 2.4). Explaining the probability output rather than the log-odds output of a logistic model yields more naturally interpretable feature attributions. Furthermore, explaining the loss of a logistic model enables debugging model performance and identification of covariate shift. A third application is neural network feature extraction, where pipelines may include transformations of the original features fed into a different model. In this setting, we demonstrate the computational tractability of G-DeepSHAP compared to model-agnostic approaches. Finally, because our approach propagates feature attributions through a series of models while satisfying efficiency at each step (Methods section Efficiency for intermediate attributions), the intermediary attributions at each part of the network can be interpreted as well. We use this to understand the

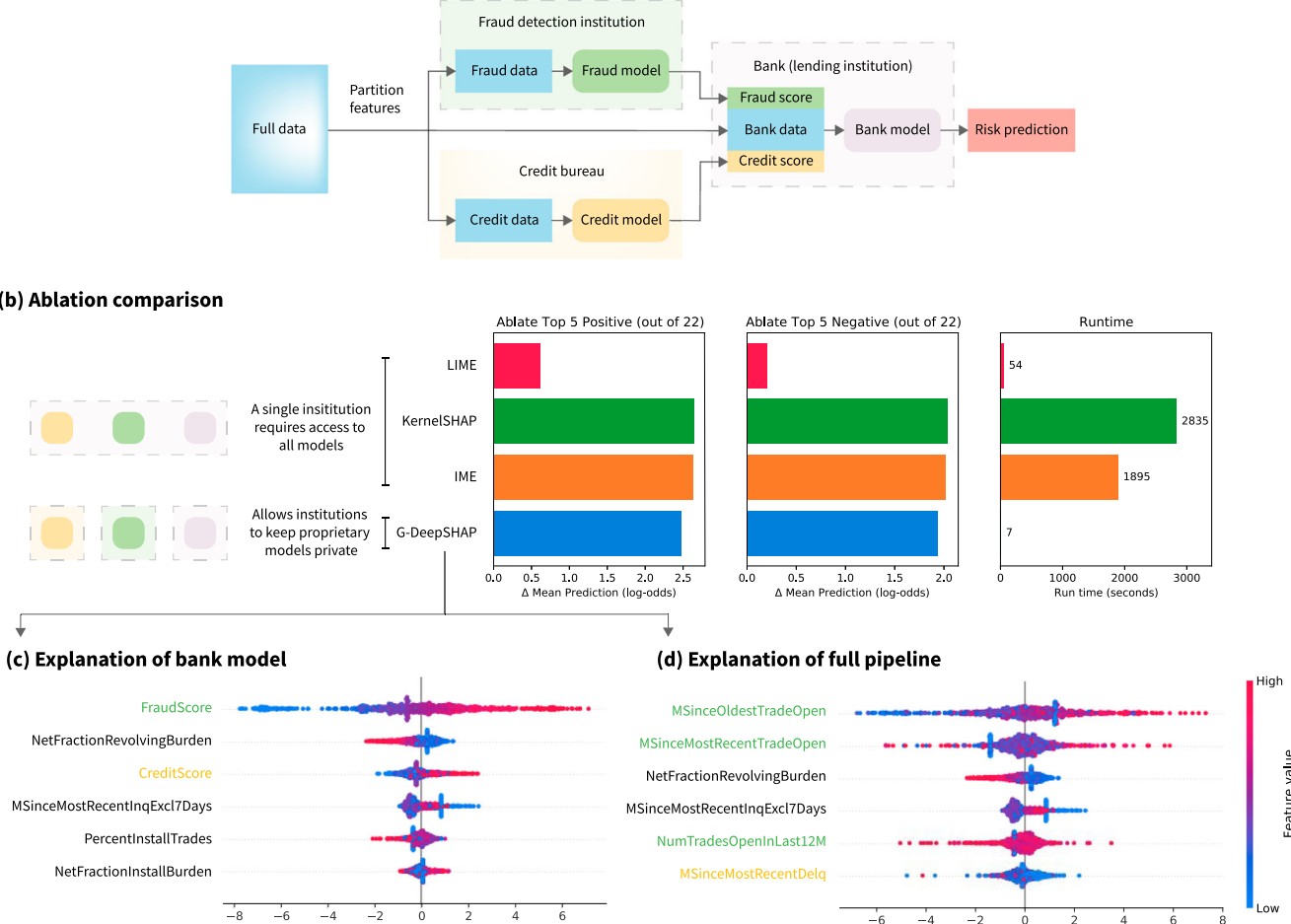

**Fig. 7 Explaining a stacked generalization pipeline of models for the HELOC data set (details in Supplementary Methods Section 1.1.7). a** A simulated model pipeline in the financial services industry. We partition the original set of features into fraud, credit, and bank features. We train a model to predict risk using fraud data and a model to predict risk using credit data. Then, we use the outputs of the fraud and credit models as scores alongside additional bank features to predict the final customer risk. **b** Ablation tests (ablating top five positive/negative features out of a total 22 features) comparing model-agnostic approaches (LIME, KernelSHAP, IME), which require access to all models in the pipeline, and G-DeepSHAP, which allows institutions to keep their proprietary models private. **c** Summary plot of the top six features the bank model uses to predict risk (TreeSHAP). **d** Summary plot of the top six features the entire pipeline uses to explain risk (G-DeepSHAP). The green features originate from the fraud data and the yellow features from the credit data. We explain 1000 randomly sampled explicands using 100 randomly sampled baselines for all attribution methods. Note that (**c**) and (**d**) show summary plots (Supplementary Methods Section 1.3.3).

importance of both consumer scores and the original features used by the consumer scores.

In consumer scoring, distributed proprietary models (i.e., models that exist in different institutions) have historically been an obstacle to transparency. This lack of transparency is particularly concerning given the prevalence of consumer scores, with some data brokers having thousands of data segments on nearly every American consumer[50]. In addition, many new consumer scores fall outside the scope of previous regulations (e.g., the Fair Credit Reporting Act and the Equal Credit Opportunity Act)[11]. In fact, these new consumer scores that depend on features correlated with protected factors (e.g., race) can reintroduce discrimination hidden behind proprietary models, which is an issue that has historically been a concern in credit scores (the oldest existing example of a consumer score)[11]. G-DeepSHAP naturally enables feature attributions in this setting and takes a significant and practical step toward increasing the transparency of consumer scores and provides a tool to help safeguard against hidden discrimination.

It should be noted that we focus specifically on evaluating G-DeepSHAP for a series of mixed model types. Previous work evaluates the rescale rule for explaining deep models, specifically. The original presentation of the rescale rule[17] demonstrates its applicability to deep networks in explaining digit classification and regulatory DNA classification. Ref. [39] show that for explaining deep networks, G-DeepSHAP, which uses multiple baselines, is a very fast yet performant approach in terms of an ablation test for explaining MNIST and CIFAR images. Although their approach, CXPlain, is comparably fast at attribution time, it has the added cost of training a separate explanation model. Finally, ref. [40] shows that many modified back propagation feature attribution techniques are independent of the parameters of later layers, with the exception of DeepLIFT. This particularly significant finding suggests that compared to most fastback propagation-based deep feature attribution approaches, approaches based on the rescale rule are not ignorant of later layers in the network.

Although G-DeepSHAP works very well for explaining a series of mixed model types in practice, an inherent limitation is that it

is not guaranteed to satisfy the desirable axioms (e.g., implementation invariance) that other feature attribution approaches satisfy (assuming exact solutions to their intractable problem formulations)[14,16,18]. This suggests that G-DeepSHAP may be more appropriate for model debugging or for identifying scientific insights that warrant deeper investigation, particularly in settings where models or the input dimension is huge and tractability is a major concern. However, for applications where high-stakes decision-making is important, it may be more appropriate to run axiomatic approaches to completion or use interpretable models[52]. Furthermore, in many real-world circumstances, such as distributed proprietary models based on credit risk scores, exact axiomatic approaches, and interpretable models are not feasible. In these cases, G-DeepSHAP represents a promising direction that allows multiple agents to collaboratively build explanations while maintaining the separation of model ownership.

## Methods

We include detailed descriptions of datasets in Supplementary Methods Section 1.1, experimental setups in Supplementary Methods Section 1.2, and feature attribution plots in Supplementary Methods Section 1.3.

**The Shapley value.** The Shapley value is a solution concept for allocating credit among players ($M = 1, \cdots, m$) in an $m$-person game. The game is fully described by a set function $v(S) : \mathcal{P}(S) \rightarrow \mathbb{R}^1$ that maps the power set of players $S \subseteq M$ to a scalar value. The Shapley value for player $i$ is the average marginal contribution of that player for all possible permutations of remaining players:

$$\phi_i(v) = \frac{1}{m!} \sum_{P \in \Sigma_{m-1}(M)} \left( v\left(S^P \cup i\right) - v\left(S^P\right) \right). \qquad (4)$$

We denote the finite symmetric group $\Sigma_{n-1}(M)$, which is the set of all possible permutations, and $S^P$ to be the set of players before player $i$ in the permutation $P$. The Shapley value is a provably unique solution under a set of axioms (Supplementary Methods Section 1.4). One axiom that we focus on in this paper is *efficiency*:

$$\sum_{i=1}^{m} \phi_i(v) = v(M) - v(\emptyset). \qquad (5)$$

*Adapting the Shapley value for feature attribution of ML models.* Unfortunately, the Shapley value cannot assign credit for an ML model ($f(x) : \mathbb{R}^m \rightarrow \mathbb{R}^1$) directly because most models require inputs with values for every feature, rather than a subset of features. Accordingly, feature attribution approaches based on the Shapley value define a new set function $v(S)$ that is a *lift* of the original model[53]. In this paper, we focus on local feature attributions, which describe a model's behavior for a single sample, called an explicand ($x^e$). A lift is defined as:

$$\mu\left(f, x^e, S\right) : \mathbb{R}^m \times 2^m \rightarrow \mathbb{R}^1. \qquad (6)$$

One common lift is the *observational conditional expectation*, where the lift is the conditional expectation of the model's output holding features in $S$ fixed to $x_S^e$ and $X$ is a multivariate random variable with joint distribution $D$:

$$\mu_D^{obs}(f, x^e, S) = \mathbb{E}_D\left[f(X)|X_S = x_S^e\right]. \qquad (7)$$

Another common lift is the interventional conditional expectation with a flat causal graph, where we "intervene" on features by breaking the dependence between features in $X_S$ and the remaining features using the causal inference do-operator[30]:

$$\mu_D^{int}(f, x^e, S) = \mathbb{E}_D\left[f(X)|do\left(X_S = x_S^e\right)\right]. \qquad (8)$$

Both approaches have tradeoffs that have been described elsewhere[13,36–38,54]. Here, we focus on the interventional approach for two primary reasons:

1. Observational Shapley values will spread credit among correlated features[54]. Although this can be desirable, it can lead to counterintuitive attributions. In particular, features that the model literally does not use to calculate its predictions will have non-zero attribution simply if they are correlated with features the model heavily depends on ref. [13]. Instead, the interventional Shapley values do a better job of identifying the features the models algebraically depend on ref. [54]. As such, the interventional Shapley values are useful for debugging bad models and drawing insights from good models. In contrast, although observational Shapley values give a view of the information content each feature has with regard to the output for optimal models[55], this is not the case for bad models. Furthermore, it can be hard to use observational Shapley values to debug bad models because it is unclear whether a feature is important because it is explicitly depended on by the

model or because it is correlated with the features the model explicitly depends on. Finally, if it is really important to spread credit using correlated features, it is possible to modify the model fitting using regularization or ensembles which will cause interventional Shapley values to naturally spread credit[54].

2. Estimating the observational conditional expectation is drastically harder than the interventional conditional expectation. This is reflected in a wide disagreement about how to estimate the observational conditional expectation[55], with approaches including empirical[13], cohort refinement[13,56,57], parametric assumptions[54,57], generative model[38], surrogate model[38], missingness during training[55], and separate models[58–60]. On the other hand, the interventional conditional expectation has one agreed-upon empirical estimation strategy[13,20]. This difficulty also reflects in the model-specific approaches, where there are exact algorithms to calculate interventional Shapley values for linear and tree models (LinearSHAP[54] and TreeSHAP[20]). In particular, TreeSHAP and G-DeepSHAP are both based on the useful property that interventional Shapley values decompose into an average of baseline Shapley values (section Interventional Shapley values baseline distribution). This benefit is crucial to the design of the generalized rescale rule.

Note that a third approach, named causal Shapley values, uses causal inference's interventional conditional expectation, but does not assume a flat causal graph[61]. Causal Shapley values require knowledge of a causal graph relating the input variables and the output. However, in general, this graph is unknown or requires substantial domain expertise. In addition, causal Shapley values are hard to estimate because they require estimating many interventional probabilities. In contrast, interventional Shapley values are a tractable way to understand model behavior.

The Shapley values computed for any lift will satisfy efficiency in terms of the lift $\mu$. However, for the interventional and observational lift described above, the Shapley value will also satisfy efficiency in terms of the model's prediction:

$$\sum_i \phi_i^{\mu_D}(f, x^e) = f(x^e) - \mathbb{E}_D[f(X)]. \qquad (9)$$

This means that attributions can naturally be understood to be in the scale of the model's predictions (e.g., log-odds or probability for binary classification).

**Interventional Shapley values baseline distribution.** We can define a single baseline lift

$$\mu_{x^b}^{int}\left(f, x^e, S\right) = \mathbb{E}_{\{x^b\}}\left[f(X)|do\left(X_S = x_S^e\right)\right] = \chi^S, \qquad (10)$$

where $\chi^S$ is a spliced sample and $\chi_i^S = x_i^e$ if $i \in S$, else $\chi_i^S = x_i^b$.

Then, we can decompose the Shapley value $\phi_i(f, x^e)$ for the interventional conditional expectation lift (eq. (8)) (henceforth referred to as the interventional Shapley value) into an average of Shapley values with single baseline lifts (proof in Supplementary Methods Section 1.6):

$$\phi_i(f, x^e, D) = \frac{1}{|D|} \sum_{x^b \in D} \underbrace{\frac{1}{m!} \sum_{P \in \Sigma_{m-1}(M)} f\left(\chi^{S^P \cup i}\right) - f\left(\chi^{S^P}\right)}_{\text{Shapley value for single baseline lift}} \qquad (11)$$

$$= \frac{1}{|D|} \sum_{x^b \in D} \phi_i\left(f, x^e, x^b\right). \qquad (12)$$

Here, $D$ is an empirical distribution with equal probability for each sample in a baseline data set. An analogous result exists for the observational conditional distribution lift using an input distribution[36]. The attributions for these single baseline games are also analogous to baseline Shapley in ref. [13].

In the original DeepLIFT paper,[17] recommends two heuristic approaches to define baseline distributions: (1) choosing a sensible single baseline and (2) averaging over multiple baselines. In addition, DeepSHAP, as previously described in ref. [16], created attributions with respect to a single baseline equal to the expected value of the inputs (Methods section Differences to previous approaches). In this paper, we show that from the perspective of Shapley values with an interventional conditional expectation lift, averaging over feature attributions computed with single baselines drawn from an empirical distribution is the correct approach. One exception to this are linear models, where taking the average as the baseline is equivalent to averaging over many single baseline feature attributions[54].

Interventional Shapley values computed with a single baseline satisfy efficiency in terms of the model's prediction:

$$\sum_i \phi_i\left(f, x^e, x^b\right) = f(x^e) - f(x^b). \qquad (13)$$

**Selecting a baseline distribution.** As in the previous section, we define a baseline distribution $D$ over which we compute Shapley values with single baseline lifts. This baseline distribution is naturally chosen to be a distribution over the training data $X^{train}$, where each sample $x^j \in \mathbb{R}^m$ has equal probability. The interpretation of this distribution is that the explicand is compared to each baseline in $D$. This means that the interventional Shapley values implicitly create attributions that explain the model's output relative to a baseline distribution.

Although the entire training distribution is a natural and interpretable choice of baseline distribution, it may be desirable to use others. To automate the process of choosing such an interpretable baseline distribution, we turn to unsupervised clustering. We utilize k-means clustering on a reduced version of the training data $(\hat{X}^{train})$ comprised of $\hat{x}^i = [x_i^j \forall i \in M_r]$ with a reduced set of features $(M_r)$. The output of the k-means clustering are clusters $C_1, \cdots, C_k$ with means $\mu_1, \cdots, \mu_k$ that minimize the following objective on the reduced training data:

$$\arg\min_{C_1, \cdots, C_k} \sum_{i=1}^{k} \sum_{\hat{x} \in C_i} \| \hat{x} - \mu_i \|^2. \tag{14}$$

Then, the cluster selected as a baseline distribution explaining an explicand $x^e$ is chosen based on:

$$\arg\min_i \| \hat{x}^e - \mu_i \|^2. \tag{15}$$

Note that in practice, it is common to use a large subsample of the full baseline distribution. The number of baseline samples can be an important parameter that can be validated by running explanations for multiple replicates and confirming consistency. We evaluate convergence in Supplementary Notes Section 2.3 and find that 1000 baselines lead to consistent attributions.

**A generalized rescale rule to explain a series of models**. We define a generalized rescale rule to explain an arbitrary series of models that propagates approximate Shapley values with an interventional conditional expectation lift for each model in the series. To describe the approach, we define a *series of models* to be a composition of functions $f_k(x) = (h_k \circ \cdots \circ h_1)(x)$, and we define intermediary models $f_i(x) = (h_i \circ \cdots \circ h_1)(x)$, $i = 1, \cdots, k$. We define the domain and codomain of each model in the series as $h_i(x) : \mathbb{R}^{m_i} \to \mathbb{R}^{o_i}$. Then, we can define the propagation for a single baseline recursively:

$$\psi^k = \hat{\phi}(h_k, x^e, x^b) \tag{16}$$

$$\psi^i = \hat{\phi}(h_i, x^e, x^b)(\psi^{i+1} \oslash (f_i(x^e) - f_i(x^b))), \ i \in 1, \cdots, k-1. \tag{17}$$

We use Hadamard division to denote an element-wise division of $\vec{a}$ by $\vec{b}$ that accommodates zero division, where if the denominator $b_i$ is 0, we set $a_i/b_i$ to 0. Additionally, $\hat{\phi}$ are an appropriate feature attribution technique that approximates interventional Shapley values while crucially satisfying efficiency for the model $h_i$ it is explaining. In this paper, we utilize DeepLIFT (rescale) for deep models, TreeSHAP for tree models, and exact interventional Shapley values for linear models. We define efficiency as $\hat{1}_{1 \times m_i} \hat{\phi}(h_i, x^e, x^b) = f_i(x^e) - f_i(x^b)$ where $\hat{1}_{a \times b}$ is a matrix of ones with shape $a \times b$ and the approximate Shapley value functions $\hat{\phi}$ return matrices in $\mathbb{R}^{(m_i \times o_i)}$. The final attributions in the original feature space are:

$$\phi_i(f_k, x^e, x^b) = \psi_i^1. \tag{18}$$

Furthermore, this approach yields intermediate attributions that serve as meaningful feature attributions. In particular, $\psi^i$ can be interpreted as the importance of the inputs to the model $(h_k \circ \cdots \circ h_i)$, where the new explicand and baseline are $(h_{i-1} \circ \cdots \circ h_1)(x^e)$ and $(h_{i-1} \circ \cdots \circ h_1)(x^b)$, respectively. This approach takes inspiration from the chain rule applied specifically for deep networks in ref. [17], but we extend it to more general classes of models.

**Efficiency for intermediate attributions**. As one might expect, each intermediate attribution $\psi^i$ satisfies efficiency:

Theorem 1: Each attribution $\psi^i \in \mathbb{R}^m, \forall i \in 1, \cdots, k$ satisfies efficiency and sums up to $f_k(x^e) - f_k(x^b)$.

Proof: We will prove by induction that

$$\hat{1}_{1 \times m_i} \psi^i = f_k(x^e) - f_k(x^b), \forall i \in 1, \cdots, k. \tag{19}$$

For simplicity of notation, denote $\hat{\phi}^i = \hat{\phi}(h^i, x^e, x^b)$.

Assumption: Each $\hat{\phi}$ satisfies efficiency

$$\hat{1}_{1 \times m_i} \hat{\phi}^i = f_i(x^e) - f_i(x^b). \tag{20}$$

Base Case: By our assumption,

$$\hat{1}_{1 \times m_k} \psi^k = f_k(x^e) - f_k(x^b). \tag{21}$$

Induction Step:

$$\psi^i = \hat{\phi}(\psi^{i+1} \oslash (f_i(x^e) - f_i(x^b))) \tag{22}$$

$$\hat{1}_{1 \times m_i} \psi^i = \hat{1}_{1 \times m_i} \hat{\phi}(\psi^{i+1} \oslash (f_i(x^e) - f_i(x^b))) \tag{23}$$

$$= (f_i(x^e) - f_i(x^b))(\psi^{i+1} \oslash (f_i(x^e) - f_i(x^b))) \tag{24}$$

$$= \hat{1}_{1 \times o_i} \psi^{i+1} \tag{25}$$

$$= \hat{1}_{1 \times m_{i+1}} \psi^{i+1} \tag{26}$$

$$= f_k(x^e) - f_k(x^b). \tag{27}$$

Conclusion: By the principle of induction, each intermediate attribution satisfies efficiency (eq. (19)).

Then, because the interventional Shapley value with a baseline distribution is the average of many single baseline attributions, it satisfies a related notion of efficiency:

$$\sum_i \phi_i(f_k, x^e) = \sum_i \sum_{x^b \in D} \phi_i(f_k, x^e, x^b) \tag{28}$$

$$= \sum_{x^b \in D} \sum_i \phi_i(f_k, x^e, x^b) \tag{29}$$

$$= \sum_{x^b \in D} f_k(x^e) - f_k(x^b) \tag{30}$$

$$= f_k(x^e) - \frac{1}{|D|} \sum_{x^b \in D} f_k(x^b). \tag{31}$$

This can be naturally interpreted as the difference between the explicand's prediction and the expected value of the function across the baseline distribution.

An additional property of the generalized rescale rule is that although it is an approximation to the interventional Shapley values in the general case, if every model in the composition is linear ($h_i(x) = \beta x$), then this propagation exactly yields the interventional Shapley values (Supplementary Methods Section 1.7).

**Connecting DeepLIFT's rules to the Shapley values**. Now we can connect the Shapley values to DeepLIFT's Rescale and RevealCancel rules. Both rules aim to satisfy an efficiency axiom (what they call summation to delta) and can be connected to an interventional conditional expectation lift with a single baseline (as in section Selecting a baseline distribution). Note that although the Rescale rule does not explicitly account for interaction effects, they can be captured in deep models, which we visualize in Supplementary Notes Section 2.8.

In fact, multi-layer perceptrons are a special case where the models in the series are nonlinearities applied to linear functions. We first represent deep models as a composition of functions $(h_1 \circ \cdots \circ h_k)(x)$. The Rescale and RevealCancel rules canonically apply to a specific class of function: $h_i(x) = (f \circ g)(x)$, where $f$ is a nonlinear function and $g$ is a linear function parameterized by $\beta \in \mathbb{R}^m$. We can interpret both rules as an approximation to interventional Shapley values based on the following definition.

**Definition 1**. A k-partition approximation to the Shapley values splits the features in $x \in \mathbb{R}^m$ into $K$ disjoint sets. Then, it exactly computes the Shapley value for each set and propagates it linearly to each component of the set.

The Rescale rule can be described as a one-partition approximation to the Interventional Shapley values for $h_i(x)$, while the RevealCancel rule can be described as a two-partition approximation that splits according to whether $\beta_i x_i > t$, where the threshold $t = 0$. This k-partition approximation lets us consider alternative variants of the Rescale and RevealCancel rules that incur exponentially larger costs in terms of $K$ and for different choices of thresholds.

**Explaining groups of input features**. Here, we further generalize the Rescale rule to support groupings of features in the input space. Having such a method can be particularly useful when explaining models with very large numbers of features that are more understandable in higher-level groups. One natural example is gene expression data, where the number of features is often extremely large.

We introduce a group rescale rule that facilitates higher-level understanding of feature attributions. It provides a natural way to impose sparsity when explaining sets of correlated features. Sparsity can be desirable when explaining a large number of features[52]. We can define a set of groups $G_1, \cdots, G_o$ whose members are the input features $x_i$. If each group is disjoint and covers the full set of features, then a natural group attribution that satisfies efficiency is the sum:

$$\phi_{G_j}^0(f, x^e) = \sum_{i \in G_j} \phi_i(f, x^e). \tag{32}$$

If the groups are not disjoint or do not cover all input features, then the above attributions do not satisfy efficiency. To address this, we define a residual group $G_R$ that covers all input features not covered by the remaining groups. Then, the new attributions are a rescaled version of eq. (32)

$$\phi_{G_j}(f, x^e) = \phi_{G_j}^0(f, x^e) \times \frac{\sum \phi_{G_j}(f, x^e)}{\sum \phi_i(f, x^e)}. \tag{33}$$

We can naturally extend this approach to accommodate nonuniform weighting of group elements, although we do not experiment with this in our paper.

**Differences to previous approaches**. In the original SHAP paper[16], they aim to calculate observational Shapley values. However, due to the difficulty in estimating observational conditional expectations, they actually calculate what is later described as interventional Shapley values[30,54].

DeepSHAP was originally introduced as an adaptation of DeepLIFT in the original SHAP paper[16] designed to make DeepLIFT closer to the interventional Shapley values. However, it is briefly and informally introduced, making it difficult to know exactly what the method entails and how it differs from DeepLIFT. DeepSHAP is the same as DeepLIFT, but with the reference (baseline) values set to the average of the baseline samples. Similar to DeepLIFT, using an average baseline also leads to bias (Supplementary Section 2.2). In comparison, DeepLIFT typically

sets the baseline to uninformative values and sets them to zeros for image data (an all-black image).

However, interventional Shapley values are equivalent to an average of baseline Shapley values, but not to baseline Shapley values with the average as a baseline. Due to this interpretation (section Interventional Shapley values baseline distribution), it is more natural to calculate G-DeepSHAP as the average of many attributions for different baselines. This in turn allows us to formulate a generalized rescale rule which allows us to propagate attributions through pipelines of a linear, tree, and deep models for which baseline Shapley values are easy to calculate.

In order to clarify the differences, we explicitly define DeepSHAP as it was originally briefly proposed in[16] and the current version we are proposing. DeepSHAP used the rescale rule with an average baseline. G-DeepSHAP uses the generalized rescale rule and group rescale rule with multiple baselines. In terms of applications, DeepSHAP only applies to deep models, whereas G-DeepSHAP applies to pipelines of a linear, tree, and deep models. Finally, the group rescale rule gives us a natural approach to group large numbers of features and thus generate attributions for a much smaller number of groups. This type of sparsity is often helpful for helping humans understand model explanations[62].

**Evaluation of explanations**. The evaluation of explanations is the topic of many papers[20,63–66]. Although there is unlikely to be a single perfect approach to evaluate local feature attributions, we can roughly separate them into two categories: qualitative and quantitative, which typically correspond to plausibility of explanations and fidelity to model behavior respectively.

Qualitative evaluations aim to ensure that relationships between features and the outcome identified by the feature attributions are correct. In general, this requires a priori knowledge of the underlying data generating mechanism. One setting in which this is possible are synthetic evaluations, where the data generating mechanism is fully known. This can be unappealing because methods that work for synthetic data may not work for real data. Instead, another approach is to externally validate with prior literature. In this case, qualitative evaluations aim to capture some underlying truth about the world that has been independently verified in diverse studies. This type of evaluation simultaneously validates the combination of the model fitting and the feature attribution itself to see whether explanations are plausible. One downside of this approach is that it can be hard to rigorously compare different explanation techniques because the evaluation is inherently qualitative. Furthermore, if the explanations find a previously unobserved relationship it can be hard to verify. For this reason, our qualitative evaluations take place in well-studied domains: mortality epidemiology, Alzheimer's and breast cancer biology, and financial risk assessment.

Then, quantitative evaluations typically aim to ensure that the feature attributions are representative of model behavior. These evaluations are dominated by feature ablation tests which aim to modify the samples in a way that should produce an expected response in the model's output. Since most local feature attributions aim to explain the model's output, it is a natural aspect of model behavior to measure. In contrast to the qualitative evaluations, quantitative evaluations are typically aimed exclusively at the feature attribution and are somewhat independent of the model being explained. These evaluations, while useful, are also imperfect, because a method that succeeds at describing model behavior perfectly will likely be far too complex to provide an explanation that humans can understand[67].

Therefore, we found it important to balance these two types of evaluations within our paper. We provide qualitative assessments for all-cause mortality, Alzheimer's, breast cancer, and loan risk performance. We additionally provide quantitative assessments for all-cause mortality, digits classification, and the loan risk performance data.

**Ablation tests**. We quantitatively evaluate our feature attribution methods with *ablation tests*[20,63]. In particular, we rely on a simple yet intuitive ablation test. For a matrix of explicands $X^e \in \mathbb{R}^{n_e, m}$, we can get attributions $\phi(f, X^e) \in \mathbb{R}^{n_e, m}$. The ablation test is defined by three parameters: (1) the feature ordering, (2) an imputation sample $x^b \in \mathbb{R}^m$, and (3) an evaluation metric. Then, the ablation test replaces features one at a time with the baseline's feature value based on the feature attributions to assess the impact on the evaluation metric. We can iteratively define the ablation test based on modified versions of the original explicands:

$$X^{e,0} = X^e \tag{34}$$

$$X^{e,k} = X^e \odot I_k(\phi) + X^b \odot \left(1 - I_k(\phi)\right), \forall k \in 1, \cdots, m. \tag{35}$$

Note that $X^b := [\underbrace{x^b \cdots x^b}_{n_e\text{elements}}]^T, I_k(\phi) := I_k(\phi(f, X^e)) = \arg\max_{k,axis=1}(\phi(f, X^e))$,

where $\arg\max_{k,axis=1}(G)$ returns an indicator matrix of the same size as $G$, 1 indicates that the element was in the maximum $k$ elements across a particular axis, and $\odot$ signifies a Hadamard product.

Then, the ablation test measures the mean model output (e.g., the predicted log-odds, predicted probability, the loss, etc.) if we ablate $k$ features to be the average

over the predictions for each ablated explicand:

$$\frac{1}{n_e} \sum_{i \in 1, \cdots, n_e} f(X_i^{e,k}). \tag{36}$$

Note that for our ablation tests we focus on either the positive or the negative elements of $\phi$, since the expected change in model output is clear if we ablate only by positive or negative attributions. Since each sample is ablated independently based on their attributions, this ablation test can be considered a summary of local ablations (Supplementary Notes Section 2.10) for many different explicands.

Ablation tests are a natural approach to test whether feature attributions are correct for a set of explicands. For feature attributions that explain the predicted log-odds, a natural choice of model output for the ablation test is the mean of the log-odds predictions. Then, as we ablate increasing numbers of features, we expect to see the model's output change. When we ablate the most positive features (according to their attributions), the mean model output should decrease substantially. As we ablate additional features, the mean model output should still decrease, but less drastically so. This implies that, for positive ablations, lower curves imply attributions that better described the model's behavior. In contrast, for negative ablations, as we ablate the most negative features, better attributions will cause the mean model output to increase rapidly and lead to higher curves. As a final note, we demonstrate that estimates of interventional Shapley values for random tree and deep models based on TreeSHAP and G-DeepSHAP respectively perform well on ablation tests (Supplementary Notes Section 2.9). This implies that our attributions closely describe the model behavior regardless of its predictive performance.

**Reporting summary**. Further information on research design is available in the Nature Research Reporting Summary linked to this article.

## Data availability

The NHANES I, NHANES 1999–2014 data are publicly available: https://wwwn.cdc.gov/nchs/nhanes/Default.aspx. The CIFAR data is publicly available: https://www.cs.toronto.edu/~kriz/cifar.html. The MNIST data is publicly available: http://yann.lecun.com/exdb/mnist/. The HELOC data set can be obtained by accepting the data set usage license: (https://community.fico.com/s/explainable-machine-learning-challenge?tabset-3158a=a4c37). METABRIC data access is restricted and requires getting an approval through the Sage Bionetworks Synapse website: https://www.synapse.org/\#\!Synapse:syn1688369 and https://www.synapse.org/#!Synapse:syn1688370. ROSMAP data access is restricted and requires getting approval through Sage Bionetworks Synapse website: https://www.synapse.org/#!Synapse:syn3219045 and is available as part of the AD Knowledge Portal[68]. Results corresponding to the figures are provided as source data with this paper. Source data are provided with this paper.

## Code availability

The code for the experiments is available here: https://github.com/suinleelab/DeepSHAP (archived at https://doi.org/10.5281/zenodo.6585445).

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

## Acknowledgements

We thank Ayse B. Dincer, Pascal Sturmfels, Joseph Janizek, and Gabe Erion for helpful discussions. This work was funded by National Science Foundation [DBI-1759487 (H.C., S.M.L., and S.-I.L.), DBI-1552309 (H.C., S.M.L., and S.-I.L.), DGE-1762114 (H.C.), and DGE-1256082 (S.M.L.)]; National Institutes of Health [R35 GM 128638 (H.C., S.M.L., and S.-I.L.), and R01 NIA AG 061132 (H.C., S.M.L., and S.-I.L.)]. ROSMAP Study data were provided by the Rush Alzheimer's Disease Center, Rush University Medical Center, Chicago. Data collection was supported through funding by NIA grants P30AG10161, R01AG15819, R01AG17917, R01AG36836, U01AG46152, U01AG46161, the Illinois Department of Public Health, and the Translational Genomics Research Institute (genomic). The METABRIC project was funded by Cancer Research UK, the British Columbia Cancer Foundation, and the Canadian Breast Cancer Foundation BC/Yukon.

## Author contributions

H.C. contributed to study design, data analysis, and manuscript preparation. S.M.L. contributed to data analysis and manuscript preparation. S.L. contributed to study design and manuscript preparation.

## Competing interests

The authors declare no competing interests.
