## [Peer Review File · Nature Communications]

REVIEWER COMMENTS

Reviewer #2 (Remarks to the Author):

Authors report a local feature attribution method, which is faster than model-agnostic approaches (e.g., kernel SHAP) and applicable to distributed series of models. DeepSHAP was reported in their NIPS 2017 paper, and they have modified the formulation. DeepSHAP is an extension of DeepLIFT algorithm to make feature attributions approximate Shapley values. DeepSHAP is influenced by the background sample distribution and authors recommend the k-Means algorithm to obtain a data set summary like in kernel SHAP.

The paper provides some novelty aspects in terms of DeepSHAP methodological extensions and proposed applications. An interesting aspect of the manuscript is how the baseline distribution can influence the local feature attributions, and the scientific question should be taken into consideration when choosing such baseline.

The analysis is extensive and complete, and the SHAP library enables reproducibility of this work. Multiple studies have reported SHAP to explain the output of ML models in different applications, so discussing some practical aspects at a higher-level might help to have more impact across fields.

1) Feature dependency or correlation

In this SHAP variant, how do correlated features influence feature attribution results? How are interaction effects accounted for? How does this relate to the feature sets and group rescale rule?

2) Explanation's validity

- How can a user know if an explanation is plausible and reproduces the complex model behaviour, especially if different baselines and initializations might lead to different explanations?
- Apart from feature ablation (which focuses on global trends on a labelled set), are there any procedures to validate whether DeepSHAP is giving a correct explanation for a given sample?
- How does model predictive performance affect model explanations? Is a minimum predictability needed to correctly generate explanations?

Additional comment:

- The last sentence of the manuscript is incomplete (pg.17, line 436).

Reviewer #3 (Remarks to the Author):

The authors present DeepSHAP, a model-agnostic technique for computing feature importance through a series of composed models with good performance in terms of computation time.

DeepSHAP is an extension of DeepLIFT, leveraging what appears to be a set of ad-hoc rules and heuristics. In this way, I would describe the novelty as somewhat limited, but would not immediately conclude that this prohibits publication alone.

My main concern, however, is that the entire approach is highly ad-hoc, both in terms of the building blocks from which the authors start as well as their new additions (e.g. interventional Shapley values, TreeSHAP & DeepLIFT, k-means clustering for baseline distribution selection). Whereas, for explainability to have an impact beyond a tool used by data scientists for iterative model development purposes, it must be highly justified and standardised. The authors do recognise this in their Discussion section when they say "This suggests that DeepSHAP may be more appropriate for model debugging ...". However, this is much weaker of a motivation than is implied in the rest of the manuscript.

The Shapley Values are an appealing paradigm for model agnostic explainability because of the mathematical control they provide: they provide a unique attribution method satisfying 4 intuitive axioms. DeepSHAP combines a series of uncontrolled biased estimates of the Shapley Values (e.g. TreeSHAP, DeepLIFT, their rescaling rule), for which the authors provide no theoretical arguments about the bias in such approximations, nor empirical comparisons to unbiased estimators (e.g. through Monte-Carlo approximation). They build on top of the interventional Shapley values, which break the correlation structure in the data (i.e. the data manifold). And, I do not think they sufficiently justify why explaining a series of models in their proposed way would actually be superior in practical contexts (e.g. consumers of the explanations being non-technical, the features used in upstream models being unknown or difficult to interpret).

As a result, I would not recommend the publication of this work in Nature Communications.

Though it does not address my main concern, and thus could not change my recommendation, I would like to commend the authors for their comprehensive empirical study, including helpful qualitative commentary alongside data sets requiring more expertise to interpret (e.g. genetics).

Reviewer #4 (Remarks to the Author):

Overall, I think the authors have written a solid manuscript. I would be willing to recommend the paper for publication after the authors have made the revisions outlined in the review that follows.

In this paper the authors present DeepSHAP, a method for explaining the predictions of complex multi-layered machine learning models using a model-agnostic local feature attribution method based on Shapley values. The Shapley values method has become popular within the explainable AI community, as it benefits from a number of game-theoretical optimality guarantees. An initial version of the proposed methodology has been introduced in the authors' previous paper as Deep SHAP (reference [17]), a combination between DeepLIFT and their SHAP method for computing Shapley values. The approach presented here is an improvement over the original method incorporating refinements and new results from the literature.

The methodology appears sound and is largely based on combining previously published results from the literature on explainable AI. The main advantage of DeepSHAP seems to lie in its flexibility. The method can provide local attributions in a number of different ways (e.g., group attributions, intermediary attributions, model loss, different baselines), it can work on mixed ML model types, and it can accommodate series of models, in which each model belongs to a separate institution. In my opinion, the authors make some good arguments and provide highly relevant examples for showing why this flexibility is necessary. Another advantage of the proposed method lies in its very quick computation time. While the showcases results are impressive, I think that the authors should stress a bit more that the reason why the method is so fast is because it only provides approximate interventional Shapley values.

I think that the authors should explicitly write down the improvements made to DeepSHAP since it was initially proposed. On lines 297-298, the authors mention an evaluation of their method by Schwab & Karen, but it is not clear to which version of the method are they referring. One difference is that compared to the original Deep SHAP method, the authors are now using the interventional Shapley values proposed by Janzing (reference [30]) instead of (conditional) Shapley values. Additionally, the authors developed a generalized DeepLIFT rescale rule with the purpose of extending the method's application to models other than neural networks. They also propose a group rescale rule that allows the method to aggregate the attributions to higher level groups of features.

I also think that the authors could make a stronger case for why they have settled for interventional Shapley values. The authors mention that this approach for computing Shapley values "is most closely related to DeepSHAP", and it seems to me that the crucial benefit of the interventional approach is their ability to decompose into a number of baseline Shapley values. I would have liked to see a comparison to the explanations produced by the original Deep SHAP method, proposed in [17]. What are the consequences of choosing this new approach when it comes to the explanations provided? From a theoretical perspective, (conditional) Shapley values and interventional Shapley values offer rather different ways of looking at the feature attribution. If the intention is to also give a causal interpretation to the explanations, then it might be worthwhile to have a look at the discussion in the paper on "Causal Shapley Values" (Heskes et al., 2020).

Perhaps the strongest aspect of the manuscript is the extensive experimental evaluation of the method. The authors have shown that the method can provide valuable insights into a number of datasets coming from unrelated domains (genetics, finance, computer vision). They compare DeepSHAP against a number of relevant competitors (IME, KernelSHAP, LIME) and show how their method can achieve a good explanatory performance through ablation experiments in a short amount of time. The practicality of the method suggests to me that this work has the potential to be a significant addition to the field.

The paper is generally well written and structured. I would like to commend the authors for their nice illustrations, which, in my opinion, greatly facilitate the understanding of the paper. However, I would like to draw their attention to the very end of the paper, right before the Acknowledgments, where it seems that they have not finished the last paragraph on ablation tests. Please make sure to fix that issue in the final version. Regarding reproducibility, another important issue is that no code has been provided. The link mentioned by the authors in Section 10 does not seem to work, at least at the time of the reviewing. Apart from that, the authors seem to have provided sufficient detail in the main paper and in the appendix to allow for the work to be reproduced.

Other comments:

- Page 2, after Equation (1): When explaining the composition of functions, it is unclear to me what the "o_i" functions are and how they are connected to the "h_i" functions.

- Page 2, footnote: I am not familiar with the term "flat causal graph", and it is not an established term to the best of my knowledge. Could the authors please provide the definition in the manuscript revision?

- Page 6, line 158: "neurodegenerative" -> "neurodegenerative"

- Page 7, line 181: "Figure 3a" -> "Figure 5a"

- Page 8, line 183: "Figure 3b" -> "Figure 5b"

- Page 11, Figure 7: "Explanation of bank model" should be subfigure (c) and "Explanation of full pipeline" should be subfigure (d).

- Page 17, Equation (35): The authors should also explain the notation for the Hadamard product. Using x^b for both the rows and for the matrix might be confusing so maybe use X^b instead?

- Page 18, line 451: The code for the experiments does not seem to be available at the link mentioned by the authors.

REVIEWER COMMENTS

Reviewer #2 (Remarks to the Author):

Authors report a local feature attribution method, which is faster than model-agnostic approaches (e.g., kernel SHAP) and applicable to distributed series of models. DeepSHAP was reported in their NIPS 2017

paper, and they have modified the formulation. DeepSHAP is an extension of DeepLIFT algorithm to make feature attributions approximate Shapley values. DeepSHAP is influenced by the background sample distribution and authors recommend the k-Means algorithm to obtain a data set summary like in kernel SHAP.

The paper provides some novelty aspects in terms of DeepSHAP methodological extensions and proposed applications. An interesting aspect of the manuscript is how the baseline distribution can influence the local feature attributions, and the scientific question should be taken into consideration when choosing such baseline.

The analysis is extensive and complete, and the SHAP library enables reproducibility of this work. Multiple studies have reported SHAP to explain the output of ML models in different applications, so discussing some practical aspects at a higher-level might help to have more impact across fields.

We thank the reviewer for their feedback on our analyses, which address fundamental questions about feature attributions, and appreciate their constructive feedback. The reviewer's suggestions and concerns are addressed below.

1) Feature dependency or correlation

In this SHAP variant, how do correlated features influence feature attribution results? How are interaction effects accounted for? How does this relate to the feature sets and group rescale rule?

We thank the reviewer for the excellent questions which have spurred us to improve our manuscript. Firstly, we would like to distinguish accounting for correlated features and capturing interaction effects and address them separately.

With respect to Shapley values, the question of *correlated features* is closely tied to the type of Shapley value feature attribution. There are two popular choices: the interventional (also known as marginal) and the observational (also known as conditional) Shapley values [14, 22, 36, 37, 53].

In our paper, we aim to estimate the interventional Shapley values because they do not spread credit among correlated features and do a better job identifying features the models algebraically depend on [52]. In contrast, observational Shapley values will spread credit among correlated features [53] and give a view of the information content each feature has with regard to the output.

Although spreading credit can be desirable, it can lead to counterintuitive attributions. For example, we can consider the scenario where we have two identical features x_1 and x_2 and we train a model that only uses feature x_1 . Interventional Shapley values will tell us that the model does not rely on feature x_2 and it will have zero attribution. In contrast, observational Shapley values will tell us that x_1 and x_2 both matter equally because they are highly correlated even though the model does not use x_2 to generate predictions [14].

Although it can still be arguable which of these approaches is preferable, it should be noted that this issue of spreading importance between correlated features is not unique to Shapley values but is rather a fundamental issue encountered in every removal-based feature attribution method [54].

For our paper, we prefer the interventional approach as it is closer to understanding the model's behavior, which can be useful for understanding good and bad models. When used to understand good models (models that the user believes to have learned meaningful patterns), it can be a tool to discover interesting non-linear associations between features and the output. When used to understand bad models it can be a tool to debug models and see where they fail. In contrast, the observational approach aims to understand the information content each feature has with regards to the output, which can be hard to understand for models with poor performance. Furthermore, if the user really wants to spread credit between correlated features, it is possible to do so during the model fitting stage (rather than the explanation stage). Using regularization or ensemble methods, interventional Shapley values will naturally spread credit as observational Shapley values do [53].

A second motivation to use interventional Shapley values is that estimating the observational conditional expectation is drastically harder than the interventional conditional expectation. This is reflected in a wide disagreement about how to estimate the observational conditional expectation [14, 37, 53, 54, 55, 56, 57, 58, 59]. On the other hand, the interventional conditional expectation has one agreed upon empirical estimation strategy [14, 21].

We have incorporated this discussion in *Section 6.1 "The Shapley value" Lines 356-375*:

Both approaches have tradeoffs that have been described elsewhere [14, 22, 36, 37, 53]. Here, we focus on the interventional approach for two primary reasons:

- 1. Observational Shapley values will spread credit among correlated features 1531. Although this can be desirable, it can lead to counterintuitive attributions. In particular, features that the model literally does not use to calculate its predictions will have non-zero attribution simply if they are correlated with features the model heavily depends on 1141. Instead, the interventional Shapley values do a better job of identifying the features the models algebraically depend on 1531. As such, the interventional Shapley values are useful for debugging bad models and drawing insights from good models. In contrast, observational Shapley values are hard to understand for models with poor performance. Finally, if it is really important to spread credit using correlated features, it is possible to modify the model fitting using regularization or ensembles which will cause interventional Shapley values to naturally spread credit 1531.**
- 2. Estimating the observational conditional expectation is drastically harder than the interventional conditional expectation. This is reflected in a wide disagreement about how to estimate the observational conditional expectation 1541, with approaches including empirical 1141, cohort refinement 114, 55, 561, parametric assumptions 153, 561, generative model 1371, surrogate model 1371, missingness during training 1541, separate models 157–591. On the other hand, the interventional conditional expectation has one agreed**

upon empirical estimation strategy [14, 21]. This difficulty also reflects in the model-specific approaches, where there are exact algorithms to calculate interventional Shapley values for linear and tree models (LinearSHAP [53] and TreeSHAP [21]). In particular, TreeSHAP and DeepSHAP are both based on the useful property that interventional Shapley values decompose into an average of baseline Shapley values (Section 6.2). This benefit is crucial to the design of the generalized rescale rule.

14. Sundararajan, M. & Najmi, A. The many Shapley values for model explanation in Proceedings of the International Conference on Machine Learning (2020), 513–523.
21. Lundberg, S. M. et al. Explainable AI for Trees: From Local Explanations to Global Understanding. CoRR abs/1905.04610. arXiv: 1905.04610. <http://arxiv.org/abs/1905.04610> (2018).
22. Merrick, L. & Taly, A. The Explanation Game: Explaining Machine Learning Models Using Shapley Values in International Cross-Domain Conference for Machine Learning and Knowledge Extraction (2020), 17–38.
36. Kumar, I. E., Venkatasubramanian, S., Scheidegger, C. & Friedler, S. Problems with Shapley-value-based explanations as feature importance measures. arXiv preprint arXiv:2002.11097 (2020).
37. Frye, C., de Mijolla, D., Cowton, L., Stanley, M. & Feige, I. Shapley-based explainability on the data manifold. arXiv preprint arXiv:2006.01272 (2020).
53. Chen, H., Janizek, J. D., Lundberg, S. & Lee, S.-I. True to the Model or True to the Data? arXiv preprint arXiv:2006.16234 (2020).
54. Covert, I., Lundberg, S. & Lee, S.-I. Explaining by removing: A unified framework for model explanation. Journal of Machine Learning Research 22, 1–90 (2021).
55. Mase, M., Owen, A. B. & Seiler, B. Explaining black box decisions by Shapley cohort refinement. arXiv preprint arXiv:1911.00467 (2019).
56. Aas, K., Jullum, M. & Loland, A. Explaining individual predictions when features are dependent: More accurate approximations to Shapley values. arXiv preprint arXiv:1903.10464 (2019).
57. Lipovetsky, S. & Conklin, M. Analysis of regression in game theory approach. Applied Stochastic Models in Business and Industry 17, 319–330 (2001).
58. trumbelj, E., Kononenko, I. & ikonja, M. R. Explaining instance classifications with interactions of subsets of feature values. Data & Knowledge Engineering 68, 886–904 (2009).
59. Williamson, B. & Feng, J. Efficient nonparametric statistical inference on population feature importance using Shapley values in International Conference on Machine Learning (2020), 10282–10291.

Then, *interaction effects* are not necessarily specific to correlated features. Instead, interaction effects are pairwise, triple, or so on interactions between the input features that affect the output (e.g., $y=x_1*x_2$, where x_1 and x_2 can be arbitrarily related statistically). As mentioned above, whether such interaction effects are captured by Shapley value feature attributions is dependent on the model being explained. For instance, if a deep model is not trained, then we do not expect to capture meaningful interaction effects with any attribution technique. However, for performant models we expect interaction effects to be useful to make good predictions.

Although DeepSHAP is not meant to provide estimates of interaction effects, we can still visualize interactions based on *vertical dispersion*. Vertical dispersion means that the same feature value has different feature importances due to interactions between features. For instance, if two men weigh 180 lbs, then the person who is 5 feet tall probably has a higher mortality risk than the other person who is 6 feet tall. This interaction effect is reflected in Shapley value feature attributions where the importance of weight to mortality risk prediction will be greater for the shorter individual, because he is overweight given his height.

In *Appendix Figure 22*, we visualize and qualitatively confirm that meaningful interactions are captured in a multi-layer perceptron explained with DeepSHAP’s rescale rule in the NHANES dataset. Firstly, we find that increasing age [69], increasing blood cadmium [70], and decreasing income ratios [71] sensibly correspond to higher mortality predictions. In the age dependence plot we can see that age greatly contributes to mortality risk. Furthermore, we find that age greatly interacts with other features. In particular, we find that although high blood cadmium increases mortality risk for younger individuals, this effect is more drastic among older individuals which is in agreement with evidence that points to an association between blood cadmium levels and Alzheimer’s disease mortality among older adults [72]. Similarly, for income ratio, we find that although high income ratios reduce mortality risk in general, the income ratio has a greater impact on mortality risk prediction for older populations. One possible hypothesis for why the effect is more drastic in older populations is that younger individuals can respond to lower incomes by increasing work effort [73], whereas older individuals may be less flexible.

In terms of the Rescale rule specifically, one mechanism to capture interactions is to encode them in later layers of the network. Then, a single node in a later hidden layer may represent the presence of an interaction. If this interaction is important to the prediction, this node will be important and this importance will be propagated back to the original features that make up the interaction.

We incorporate a reference in the main text (*Methods Section 6.6 “Connecting DeepLIFT’s rules to the Shapley values”*) to a new *Appendix Section A.5.6 “Visualizing interaction effects with DeepSHAP”* detailing the above discussion alongside *Appendix Figure 22* below.

Figure 22: Plotting dependence plots to visualize interactions for an MLP with two 128-node hidden layers and dropout layers trained to predict all-cause mortality in the NHANES dataset (test ROC 0.838).

- 69. Kesteloot, H. & Huang, X. On the relationship between human all-cause mortality and age. *European journal of epidemiology* 18, 503–511 (2003).
- 70. Nawrot, T. S. et al. Cadmium-related mortality and long-term secular trends in the cadmium body burden of an environmentally exposed population. *Environmental health perspectives* 116, 1620–1628 (2008).
- 71. Sabanayagam, C. & Shankar, A. Income is a stronger predictor of mortality than education in a national sample of US adults. *Journal of health, population, and nutrition* 30, 82 (2012).
- 72. Min, J.-y. & Min, K.-b. Blood cadmium levels and Alzheimer’s disease mortality risk in older US adults. *Environmental Health* 15, 1–6 (2016).
- 73. Snyder, S. E. & Evans, W. N. The effect of income on mortality: evidence from the social security notch. *The review of economics and statistics* 88, 482–495 (2006).

Finally, the group rescale rule is indeed related to feature correlation. Oftentimes, the sets of features are chosen based on a fundamental relationship (e.g., gene sets). As such, the group rescale rule can be viewed as a natural way to impose sparsity when explaining sets of correlated features. Sparsity can be a desiderata of explanation techniques when there are many features [51]. We include this briefly in *Section 6.7 “Explaining groups of input features” Lines 469-471.*

We introduce a group rescale rule that facilitates higher level understanding of feature attributions. **It provides a natural way to impose sparsity when explaining sets of correlated features. Sparsity can be desirable when explaining a large number of features [51].**

51. Rudin, C. Stop explaining black box machine learning models for high stakes decisions and use interpretable models instead. *Nature Machine Intelligence* 1, 206–215 (2019).

2) *Explanation’s validity*

- How can a user know if an explanation is plausible and reproduces the complex model behaviour, especially if different baselines and initializations might lead to different explanations?

We thank the reviewer for the excellent question. The evaluation of explanations is the topic of many papers [21, 62-65]. Although there is unlikely to be a single perfect approach to evaluate local feature attributions, we can roughly separate them into two categories: qualitative and quantitative, which typically correspond to plausibility of explanations and fidelity to model behavior respectively.

Qualitative evaluations aim to ensure that relationships between features and the outcome identified by the feature attributions are correct. In general, this requires a priori knowledge of the underlying data generating mechanism. One setting in which this is possible are synthetic evaluations, where the data generating mechanism is fully known. This can be unappealing because methods that work for synthetic data may not work for real data. Instead, another appealing approach to qualitatively evaluate feature attributions is via external validation with prior literature. In this case, qualitative evaluations aim to capture some underlying truth about the world that has been independently verified in diverse studies. This type of evaluation simultaneously validates the combination of the model fitting and the feature attribution to see whether explanations agree with prior knowledge. One downside of this approach is that it is hard to compare different explanation techniques because the evaluation is inherently qualitative. Furthermore, if the explanations find a previously unobserved relationship it can be hard to verify. For this reason, our qualitative evaluations evaluate the sensibility of DeepSHAP attributions in well-studied domains: mortality prediction, Alzheimer’s and breast cancer biology, and financial risk assessment.

Then, quantitative evaluations typically aim to ensure that the feature attributions are representative of model behavior and are often used to compare explanation techniques. These evaluations are dominated by feature ablation tests [21] which aim to modify the samples in a way that should produce an expected response in the model’s output. Since most local feature attributions aim to explain the model’s output, it is a natural aspect of model behavior to measure. In contrast to the qualitative evaluations, quantitative evaluations are typically aimed exclusively at the feature attribution and can be independent of the model being explained. These evaluations, while useful, are also imperfect, because a method that succeeds at

describing model behavior perfectly will likely be far too complex to provide an explanation that humans can understand.

Therefore, we believe a balance of these two types of evaluations provides strong evidence for explanation plausibility. We provide qualitative assessments for all-cause mortality, Alzheimer’s, breast cancer, and loan risk performance to ensure that our explanations are plausible. We additionally provide quantitative assessments for all-cause mortality, digits classification, and the loan risk performance data to ensure our explanations capture model behavior [67].

We have incorporated the above discussion into a new *Methods Section 6.9 “Evaluation of explanations” Lines 501-529.*

21. Lundberg, S. M. et al. Explainable AI for Trees: From Local Explanations to Global Understanding. CoRR abs/1905.04610. arXiv: 1905.04610. <http://arxiv.org/abs/1905.04610> (2018).
62. Hooker, S., Erhan, D., Kindermans, P.-J. & Kim, B. A benchmark for interpretability methods in deep neural networks in *Advances in Neural Information Processing Systems* (2019), 9737–9748.
63. Doshi-Velez, F. & Kim, B. Towards a rigorous science of interpretable machine learning. arXiv preprint arXiv:1702.08608 (2017).
64. Murdoch, W. J., Singh, C., Kumbier, K., Abbasi-Asl, R. & Yu, B. Definitions, methods, and applications in interpretable machine learning. *Proceedings of the National Academy of Sciences* 116, 22071–22080 (2019).
65. Adebayo, J. et al. Sanity checks for saliency maps. arXiv preprint arXiv:1810.03292 (2018).
67. Weld, D. S. & Bansal, G. The challenge of crafting intelligible intelligence. *Communications of the ACM* 62, 70–79 (2019).

In response to the question about *different baselines*, it is standard to estimate interventional Shapley values using a subsample of baselines. In our experiments we typically use 1000 random baseline samples from the general population (the full training data). Although it is common to use a subsample of baselines because it is faster, it can lead to variability in the resultant attributions. In this section, we design an experiment to test how variable our attributions are based on the size of the baseline set. We utilize the NHANES 1994-2014 dataset (Appendix Section A.1.2), which serves as a suitable testing ground with a large number of samples (35,854) and a large number of features (153). In *Figures 11 and 12* we find that the attributions generate consistent attributions globally and locally respectively even for different randomly sampled sets of baselines. We incorporate these figures in a new *Appendix Section A.4.3 “Variability due to baselines” Lines 934-950.*

Figure 11: Global variability due to baselines. We replicate attributions on the entire test set ($n=7034$) using three different baseline sets comprising 1000 random samples from the training set. In (a) we show the attributions for a tree (XGB) model and in (b) we show the attributions for a deep (MLP) model. Note that (a) and (b) show dependence plots (Appendix Section A.3.2).

More generally, the number of baseline samples can be an important hyperparameter that users can verify by simply running their explanations multiple times and confirming consistency. Note that these baselines are not inherent to DeepSHAP and are important parameters for many methods that rely on Shapley values. We add this recommendation in the paper in *Methods Section 6.3 “Selecting a baseline distribution” Lines 414-417*:

Note that in practice, it is common to use a large subsample of the full baseline distribution. The number of baseline samples can be an important parameter that can be validated by running

explanations for multiple replicates and confirming consistency. We evaluate convergence in Appendix Section A.4.1 and find that 1000 baselines lead to fairly consistent attributions.

In terms of baselines sampled from specific subpopulations, we would expect similar convergence in the number of baselines. In terms of knowing whether these attributions are correct, we can only rely on the aforementioned quantitative and qualitative evaluations. We performed quantitative evaluations in *Figure 3* by ablating with an older male imputation sample. Doing so evaluates how well we capture model behavior relative to an older male subpopulation.

Finally, the fact that *different initializations* can lead to different explanations can actually be a good thing. As we discussed, there are two main goals of model explanations: (1) to learn about the model’s behavior and use it to diagnose unexpected behavior or (2) to learn a pattern in the data which hopefully reflects a true relationship in the world.

In terms of learning the model’s behavior, having different explanations for different initializations is good, because explanations that are faithful to the model behavior should change for different models. To test this in an extreme setting, we demonstrate that our explanations are faithful to the model even for models with random predictive performance using ablation tests which are targeted towards evaluating this faithfulness. We add this result in a new *Appendix Section A.5.7 “Explanations are robust to predictive performance” Lines 1061-1071:*

One of the main goals of model explanations is to learn about the model’s behavior and use it to diagnose unexpected behavior. Interventional Shapley values provide a close description of model behavior and thus can be used regardless of the quality of the trained model. We designed a simple experiment on the NHANES 1999-2014 dataset to illustrate this for two “random” models: a tree (XGB) model and a deep (MLP) model. Prior to training each model we randomly shuffle the training labels. This results in models with random test performance (ROC ~0.5). Then, we use ablation tests to evaluate the faithfulness of the attributions to the model behavior in Appendix Figure 23. We find that ablating by the attributions (exact interventional Shapley values for the tree model and approximate for the deep model) drastically changes mean model outputs which indicates they are good descriptions of the model’s behavior.

Figure 23: Positive and negative ablation tests based on attributions for (a) a random tree (XGB) model (test ROC 0.416) explained with TreeSHAP and (b) a random deep (MLP) model (test ROC 0.481) explained with DeepSHAP.

In terms of learning true patterns in the data, having different initializations lead to different explanations may be undesirable. However, this problem may be due to the Rashomon effect, where there can be multiple accurate models that rely on different input patterns [52]. In our opinion, this issue is not the responsibility of the explanation method. Instead, this should be dealt with when fitting the model through careful validation and/or by using ensembles of disagreeing models. Once the user is confident in their model, a feature attribution technique that is faithful to model behavior such as interventional Shapley values can be useful for surfacing meaningful patterns.

52. Rudin, C. Stop explaining black box machine learning models for high stakes decisions and use interpretable models instead. *Nature Machine Intelligence* 1, 206–215 (2019).

- Apart from feature ablation (which focuses on global trends on a labelled set), are there any procedures to validate whether DeepSHAP is giving a correct explanation for a given sample?

We thank the reviewer for the excellent question. We are slightly unsure if the reviewer is asking about alternatives to ablation tests or local ablations which are targeted to evaluate explanations with single samples. To be safe, we address both points below.

In terms of alternatives to ablation tests, it can be hard to define the “correctness” of an explanation. In our opinion, the best explanation is one that closely describes the model’s behavior. Then, these explanations can be used to debug poorly performing models and find interesting patterns from very accurate models. As the reviewer points out, to evaluate how well explanations describe model behavior, we use ablation tests. Ablation tests are widely used to evaluate how well explanations describe model behavior and are the only quantitative methodology we are aware of.

In terms of local ablations, it should be noted that although the ablation tests we use evaluate global trends by ablating many samples and averaging the mean predictions, they are implicitly validating local trends for many samples. These local ablations can be used to validate that DeepSHAP is giving correct explanations for single samples. To better clarify local ablations, we include an example in a new *Appendix Section A.5.8* entitled “*Example of a local ablation*” Lines 1072-1078:

In our ablation tests we primarily report global metrics aggregated over many samples (e.g., mean loss or mean model output). It is worth noting that the global ablation tests are aggregates of local ablations. We show an example of one such local ablation for the XGB model (used in Figure 5c and 5d) trained on the NHANES 1999-2014 data and explained using TreeSHAP in Figure 24. Local ablation tests serve to assess the quality of the attributions in terms of capturing model behavior for a single sample, whereas global attributions summarize the performance of local ablations across many samples.

We include a reference to this section in *Methods Section 6.10 "Ablation tests" Lines 534-536*:

Note that for our ablation tests we focus on either the positive or the negative elements of ϕ , since the expected change in model output is clear if we ablate only by positive or negative attributions. **Since each sample is ablated independently based on their attributions, this ablation test can be considered a summary of local ablations (Appendix Section A.5.5) for many different explicands.**

- How does model predictive performance affect model explanations? Is a minimum predictability needed to correctly generate explanations?

We thank the reviewer for the excellent question. We believe this is related to a previous question about different randomly initialized models. To briefly reiterate, we demonstrate that our explanations are faithful to the model (based on ablation tests) even for the extreme case of models with random predictive performance. We add this result in *Appendix Section A.5.7 “Explanations are robust to predictive performance” Lines 1061-1071*:

One of the main goals of model explanations is to learn about the model’s behavior and use it to diagnose unexpected behavior. Interventional Shapley values provide a close description of model behavior and thus can be used regardless of the quality of the trained model. We design a simple experiment on the NHANES 1999-2014 dataset to illustrate this for two “random” models: a tree (XGB) model and a deep (MLP) model. Prior to training each model we randomly shuffle the training labels. This results in models with random test performance (ROC ~0.5). Then, we use ablation tests to evaluate the faithfulness of the attributions to the model behavior in Appendix Figure 23. We find that ablating by the attributions (exact interventional Shapley values for the tree model and approximate for the deep model) drastically changes mean model outputs which indicates they are good descriptions of the model’s behavior.

Figure 23: Positive and negative ablation tests based on attributions for (a) a random tree (XGB) model (test ROC 0.416) explained with TreeSHAP and (b) a random deep (MLP) model (test ROC 0.481) explained with DeepSHAP. Attributions were computed with 1000 randomly sampled explicands explained using 1000 randomly sampled baselines.

One final note is that Shapley value feature attributions generally do not make assumptions on the model’s performance. The exception to this is that the observational Shapley values have nice information theoretic connections, but only under the assumptions that the model is a global optimum and the conditional distributions are correctly modeled [54]. These information theoretic ties are perhaps one reason to prefer observational Shapley values, but the requisite assumptions typically do not hold in practice.

54. Covert, I., Lundberg, S. & Lee, S.-I. Explaining by removing: A unified framework for model explanation. *Journal of Machine Learning Research* 22, 1–90 (2021).

Additional comment:

- The last sentence of the manuscript is incomplete (pg.17, line 436).

We thank the reviewer for catching this sentence. We have completed the sentence:

In contrast, for negative ablations, **as we ablate the most negative features, better attributions will cause the mean model output to increase rapidly and lead to higher curves.**

Reviewer #3 (Remarks to the Author):

The authors present DeepSHAP, a model-agnostic technique for computing feature importance through a series of composed models with good performance in terms of computation time.

We thank the reviewer for their detailed feedback. We greatly appreciate the reviewer’s suggestion to include quantitative comparisons to unbiased estimators of the interventional Shapley values. We failed to clarify this important point in the main text, but would like to note that **two of the methods** (KernelSHAP [17] and IME [15]) **we compared to** (Figures 5d, 6a, 6b, and 7b) **actually are unbiased estimators of the interventional Shapley values** [15, 22]. We add this to the main text *Section 1 “Introduction” Lines 63-65.*

In this paper, we present DeepSHAP – a local feature attribution method that is faster than model-agnostic methods and can explain complex series of models that pre-existing model-specific methods cannot. **We compare to extremely popular model-agnostic methods including KernelSHAP and IME which are unbiased stochastic estimators for the Shapley value [15, 17, 22].** DeepSHAP is based on connections to the Shapley value, a concept from game theory that satisfies many desirable axioms.

15. Strumbelj, E. & Kononenko, I. An efficient explanation of individual classifications using game theory. *The Journal of Machine Learning Research* 11, 1–18 (2010).

17. Lundberg, S. M. & Lee, S.-I. A unified approach to interpreting model predictions in *Advances in Neural Information Processing Systems* (2017), 4765–4774.

22. Covert, I. & Lee, S.-I. Improving KernelSHAP: Practical Shapley value estimation using linear regression in *International Conference on Artificial Intelligence and Statistics* (2021), 3457–3465.

We clarify this in more detail below and address the remaining suggestions and concerns raised by the reviewer.

DeepSHAP is an extension of DeepLIFT, leveraging what appears to be a set of ad-hoc rules and heuristics. In this way, I would describe the novelty as somewhat limited, but would not immediately conclude that this prohibits publication alone.

My main concern, however, is that the entire approach is highly ad-hoc, both in terms of the building blocks from which the authors start as well as their new additions (e.g. interventional Shapley values, TreeSHAP & DeepLIFT, k-means clustering for baseline distribution selection).

We thank the reviewer for addressing this point of clarification, which has encouraged us to improve our descriptions of these building blocks. We would like to clarify that two of the aforementioned building blocks are not ad-hoc: interventional Shapley values and TreeSHAP.

Although the original SHAP paper defines observational Shapley values, they actually calculate interventional Shapley values, due to the difficulty in estimating observational conditional expectations. Since then, interventional Shapley values have since been advocated for based on their theoretical properties [14,31] and are the default explanation provided by the popular SHAP package (<https://github.com/slundberg/shap>). We clarify this in a new Methods *Section 6.8 “Differences to previous approaches” Lines 478-480*.

14. Sundararajan, M. & Najmi, A. The many Shapley values for model explanation in Proceedings of the International Conference on Machine Learning (2020), 513–523.

In the original SHAP paper [17], they aim to calculate observational Shapley values. However, due to the difficulty in estimating observational conditional expectations, they actually calculate what is later described as interventional Shapley values [31, 54].

17. Lundberg, S. M. & Lee, S.-I. A unified approach to interpreting model predictions in Advances in Neural Information Processing Systems (2017), 4765–4774.

31. Janzing, D., Minorics, L. & Blöbaum, P. Feature relevance quantification in explainable AI: A causality problem. arXiv preprint arXiv:1910.13413 (2019).

54. Chen, H., Janizek, J. D., Lundberg, S. & Lee, S.-I. True to the Model or True to the Data? arXiv preprint arXiv:2006.16234 (2020).

Furthermore, interventional Shapley values can be preferable to their primary alternative, observational Shapley values. In our paper, we aim to estimate the interventional Shapley values because they do not spread credit among correlated features and do a better job identifying features the models algebraically depend on [54]. In contrast, observational Shapley values will spread credit among correlated features [55] and give a view of the information content each feature has with regard to the output.

Although spreading credit can be desirable, it can lead to counterintuitive attributions. For example, we can consider the scenario where we have two identical features x_1 and x_2 and we train a model that only uses feature x_1 . Interventional Shapley values will tell us that the model does not rely on feature x_2 and it will have zero attribution. In contrast, observational Shapley values will tell us that x_1 and x_2 both matter equally because they are highly correlated even though the model does not use x_2 to generate predictions [14].

Although it can still be arguable which of these approaches is preferable, it should be noted that this issue of spreading importance between correlated features is not unique to Shapley values but is rather a fundamental issue encountered in every removal-based feature attribution method [55].

For our paper, we prefer the interventional approach as it is closer to understanding the model’s behavior, which can be useful for understanding good and bad models. When used to understand good models (models that the user believes to have learned meaningful patterns), it can be a tool to discover interesting

non-linear associations between features and the output. When used to understand bad models it can be a tool to debug models and see where they fail. In contrast, the observational approach aims to understand the information content each feature has with regards to the output, which can be hard to understand for models with poor performance. Furthermore, if the user really wants to spread credit between correlated features, it is possible to do so during the model fitting stage (rather than the explanation stage). Using regularization or ensemble methods, interventional Shapley values will naturally spread credit as observational Shapley values do [54].

A second motivation to use interventional Shapley values is that estimating the observational conditional expectation is drastically harder than the interventional conditional expectation. This is reflected in a wide disagreement about how to estimate the observational conditional expectation [14, 38, 54, 55, 56, 57, 58, 59, 60]. On the other hand, the interventional conditional expectation has one agreed upon empirical estimation strategy [14, 21].

We have incorporated this discussion in *Section 6.1 “The Shapley value” Lines 356-375*:

- 1. Observational Shapley values will spread credit among correlated features 1541. Although this can be desirable, it can lead to counterintuitive attributions. In particular, features that the model literally does not use to calculate its predictions will have non-zero attribution simply if they are correlated with features the model heavily depends on 1141. Instead, the interventional Shapley values do a better job of identifying the features the models algebraically depend on 1541. As such, the interventional Shapley values are useful for debugging bad models and drawing insights from good models. In contrast, observational Shapley values are hard to understand for models with poor performance. Finally, if it is really important to spread credit using correlated features, it is possible to modify the model fitting using regularization or ensembles which will cause interventional Shapley values to naturally spread credit 1541.**
- 2. Estimating the observational conditional expectation is drastically harder than the interventional conditional expectation. This is reflected in a wide disagreement about how to estimate the observational conditional expectation 1551, with approaches including empirical 1141, cohort refinement 114, 56, 571, parametric assumptions 154, 571, generative model 1381, surrogate model 1381, missingness during training 1551, and separate models 158–601. On the other hand, the interventional conditional expectation has one agreed upon empirical estimation strategy 114, 211. This difficulty also reflects in the model-specific approaches, where there are exact algorithms to calculate interventional Shapley values for linear and tree models (LinearSHAP 1541 and TreeSHAP 1211). In particular, TreeSHAP and DeepSHAP are both based on the useful property that interventional Shapley values decompose into an average of baseline Shapley values (Section 6.2). This benefit is crucial to the design of the generalized rescale rule.**

14. Sundararajan, M. & Najmi, A. The many Shapley values for model explanation in Proceedings of the International Conference on Machine Learning (2020), 513–523.

21. Lundberg, S. M. et al. Explainable AI for Trees: From Local Explanations to Global Understanding. CoRR abs/1905.04610. arXiv: 1905.04610. <http://arxiv.org/abs/1905.04610> (2018).

38. Frye, C., de Mijolla, D., Cowton, L., Stanley, M. & Feige, I. Shapley-based explainability on the data manifold. arXiv preprint arXiv:2006.01272 (2020).
54. Chen, H., Janizek, J. D., Lundberg, S. & Lee, S.-I. True to the Model or True to the Data? arXiv preprint arXiv:2006.16234 (2020).
55. Covert, I., Lundberg, S. & Lee, S.-I. Explaining by removing: A unified framework for model explanation. *Journal of Machine Learning Research* 22, 1–90 (2021).
56. Mase, M., Owen, A. B. & Seiler, B. Explaining black box decisions by Shapley cohort refinement. arXiv preprint arXiv:1911.00467 (2019).
57. Aas, K., Jullum, M. & Loland, A. Explaining individual predictions when features are dependent: More accurate approximations to Shapley values. arXiv preprint arXiv:1903.10464 (2019).
58. Lipovetsky, S. & Conklin, M. Analysis of regression in game theory approach. *Applied Stochastic Models in Business and Industry* 17, 319–330 (2001).
59. Štrumbelj, E., Kononenko, I. & Šikonja, M. R. Explaining instance classifications with interactions of subsets of feature values. *Data & Knowledge Engineering* 68, 886–904 (2009).
60. Williamson, B. & Feng, J. Efficient nonparametric statistical inference on population feature importance using Shapley values in *International Conference on Machine Learning* (2020), 10282–10291.

As mentioned above, TreeSHAP is not an ad-hoc approach. Although estimating Shapley values is NP-hard in general, for linear and tree models, it is possible to exactly estimate interventional Shapley values tractably. In particular, we utilize a variant of TreeSHAP that exactly computes the interventional Shapley values by decomposing the interventional Shapley values into an average of baseline Shapley values [21].

21. Lundberg, S. M. et al. Explainable AI for Trees: From Local Explanations to Global Understanding. CoRR abs/1905.04610. arXiv: 1905.04610. <http://arxiv.org/abs/1905.04610> (2018).

Although some of the remaining building blocks can be viewed as ad-hoc, we would like to note that many explanation approaches are also ad-hoc, but this does not necessarily mean they are not useful. A few popular examples include layer-wise relevance propagation (LRP) [1], GradCAM [2], and DeepLIFT [3]. LRP and DeepLIFT's primary motivations are to utilize heuristic techniques at each layer and ensure a form of efficiency such that the final attributions sum up to the explicand's prediction, which is the same motivation we follow with the generalized rescale rule. For DeepLIFT in particular, despite its potentially ad-hoc nature, it has been used to identify meaningful scientific insights verified by domain experts in terms of identifying relevant motor cortices in electroencephalogram signals [4] and discovering transcription factor binding motifs in genomic data [5]. These successful applications are evidence that ad-hoc techniques can be effective at identifying meaningful patterns in performant models.

[1] Bach, Sebastian, et al. "On pixel-wise explanations for non-linear classifier decisions by layer-wise relevance propagation." *PloS one* 10.7 (2015): e0130140.

[2] Selvaraju, Ramprasaath R., et al. "Grad-cam: Visual explanations from deep networks via gradient-based localization." *Proceedings of the IEEE international conference on computer vision*. 2017.

[3] Shrikumar, Avanti, Peyton Greenside, and Anshul Kundaje. "Learning important features through propagating activation differences." *International conference on machine learning*. PMLR, 2017.

[4] Lawhern, Vernon J., et al. "EEGNet: a compact convolutional neural network for EEG-based brain-computer interfaces." *Journal of neural engineering* 15.5 (2018): 056013.

[5] Avsec, Žiga, et al. "Base-resolution models of transcription-factor binding reveal soft motif syntax." *Nature Genetics* 53.3 (2021): 354-366.

Whereas, for explainability to have an impact beyond a tool used by data scientists for iterative model development purposes, it must be highly justified and standardised. The authors do recognise this in their Discussion section when they say "This suggests that DeepSHAP may be more appropriate for model debugging ... ". However, this is much weaker of a motivation than is implied in the rest of the manuscript.

Despite successful applications of adhoc explanation techniques, we do strongly agree with the reviewer that explainability techniques should be highly justified. In fact, some recent research has pointed out that many saliency methods fail sanity checks based on deep model randomization [6,7]. In particular, [7] found that many modified backpropagation algorithms (Deep Taylor Decomposition, LRP, Guided BP, etc.), with the exception of DeepLIFT, are independent of the parameters of later layers.

Although Deep Taylor Decomposition is justified from a theoretical standpoint, it still fails a simple sanity check. One possible reason such approaches are popular despite failing to identify meaningful features could be that they often rely on qualitative evaluations via visual inspection of saliency maps [1,2]. Note that the original LRP paper does include quantitative evaluations, but primarily evaluates the effect of ablations on model loss (rather than model output) [1]. Both of these evaluations would give good performance to edge detectors.

Instead, in order to justify DeepSHAP, we focus on a balance of qualitative and quantitative evaluations, where our quantitative evaluations directly test whether hiding (ablating) features with positive attribution by replacing them with their mean pushes the model's output to be less positive and vice versa for negative attributions. For these tests, edge detectors would fail, because they cannot distinguish whether features influence the model's output positively or negatively. Furthermore, edge detectors would not work for the tabular datasets we consider. In order to more directly test the attributions, we focus on ablation tests that directly measure the model behavior [8] we are trying to capture: loss ablations for loss attributions and output ablations for output attributions (Figure 5d).

Another reason that we focus on empirical evaluations is that even axiomatic approaches such as SHAP [9] and integrated gradients (IG) [10] are not faultless. In fact, one major disadvantage of these approaches is that they are impossible to compute exactly in most practical settings. The Shapley values are NP-hard to compute in general [11] and IG requires integration along a path. As a consequence, even though these methods are highly justified, their estimators can be slow and may have high error in practice.

For the interventional Shapley values in particular, a few unbiased estimators include KernelSHAP [9] and IME [12]. These estimators introduce an additional parameter which is the number of coalitions (subsets of features) for which they evaluate the model. This means that although the interventional Shapley values will satisfy desirable axioms and perform very well on ablation tests, their estimates, which use a subsample of the total number of coalitions, may not. We compare to both approaches (KernelExplainer and SamplingExplainer from the SHAP package) with their default number of coalitions for NHANES, MNIST, and HELOC in Figures 5d, 6a, 6b, and 7b. Furthermore, these stochastic estimators can be extremely expensive which is compounded by the difficulty of detecting convergence. Given the combination of performance on ablation tests and computational efficiency, we find that DeepSHAP explanations can be extremely practical and effective despite their bias.

Finally, given that DeepSHAP provides close descriptions of model behavior, regardless of model performance (*Appendix Section A.5.7 “Explanations are robust to predictive performance”*), we believe DeepSHAP can not only be a useful debugging tool for poorly performing models but also a way to draw scientific insights from generalizable models.

- [1]Bach, Sebastian, et al. "On pixel-wise explanations for non-linear classifier decisions by layer-wise relevance propagation." *PloS one* 10.7 (2015): e0130140.
- [2]Selvaraju, Ramprasaath R., et al. "Grad-cam: Visual explanations from deep networks via gradient-based localization." *Proceedings of the IEEE international conference on computer vision*. 2017.
- [3]Shrikumar, Avanti, Peyton Greenside, and Anshul Kundaje. "Learning important features through propagating activation differences." *International conference on machine learning*. PMLR, 2017.
- [4]Lawhern, Vernon J., et al. "EEGNet: a compact convolutional neural network for EEG-based brain–computer interfaces." *Journal of neural engineering* 15.5 (2018): 056013.
- [5]Avsec, Žiga, et al. "Base-resolution models of transcription-factor binding reveal soft motif syntax." *Nature Genetics* 53.3 (2021): 354-366.
- [6]Adebayo, Julius, et al. "Sanity checks for saliency maps." *Advances in neural information processing systems* 31 (2018).
- [7]Sixt, Leon, Maximilian Granz, and Tim Landgraf. "When explanations lie: Why many modified bp attributions fail." *International Conference on Machine Learning*. PMLR, 2020.
- [8]Covert, Ian, Scott Lundberg, and Su-In Lee. "Explaining by removing: A unified framework for model explanation." *Journal of Machine Learning Research* 22.209 (2021): 1-90.
- [9]Lundberg, S. M. & Lee, S.-I. A unified approach to interpreting model predictions in *Advances in Neural Information Processing Systems* (2017), 4765–4774.
- [10] Sundararajan, M., Taly, A. & Yan, Q. Axiomatic attribution for deep networks. *arXiv preprint arXiv:1703.01365* (2017).
- [11] Deng, Xiaotie, and Christos H. Papadimitriou. "On the complexity of cooperative solution concepts." *Mathematics of operations research* 19.2 (1994): 257-266.
- [12] Strumbelj, E. & Kononenko, I. An efficient explanation of individual classifications using game theory. *The Journal of Machine Learning Research* 11, 1–18 (2010).

The Shapley Values are an appealing paradigm for model agnostic explainability because of the mathematical control they provide: they provide a unique attribution method satisfying 4 intuitive axioms. DeepSHAP combines a series of uncontrolled biased estimates of the Shapley Values (e.g. TreeSHAP, DeepLIFT, their rescaling rule), for which the authors provide no theoretical arguments about the bias in such approximations, nor empirical comparisons to unbiased estimators (e.g. through Monte-Carlo approximation).

We thank the reviewer for the insightful question. It may be difficult to provide theoretical arguments about the bias in these approximations because there can be a mix of model types: linear, tree, and deep. Instead, we focus on a balance of qualitative and quantitative empirical evaluations. As mentioned previously we do have quantitative comparisons of DeepSHAP to unbiased estimators using a fixed number of coalitions in the main text experiments (the default settings in `SamplingExplainer` and `KernelExplainer` use $2^d + 2048$ coalitions, where d is the number of features).

However, to further evaluate the bias of our approximations and the variance of these unbiased estimators, we introduce additional quantitative comparisons in three settings: NHANES loss explanations, MNIST feature extraction, and HELOC model stacking. To evaluate how the number of coalitions affects the IME and KernelSHAP estimates, we re-run them for 10 iterations at increasing numbers of coalitions. In Figures 25a, 26a, and 27a, we visualize the standard deviation of the estimates which exponentially

decays with increasing numbers of coalitions. Although it is clear that more coalitions leads to lower standard deviation, having more coalitions is also very expensive. Therefore, we may wish to evaluate how this standard deviation impacts how well the attributions describe model behavior.

To assess this, we visualize ablation tests for DeepSHAP in addition to KernelSHAP and IME with varying numbers of coalitions in Figures 25b, 25c, 26b, and 27b. In general, we find that KernelSHAP and IME are orders of magnitude slower than DeepSHAP even with very small numbers of coalitions (Figure 28). Furthermore, we find that for small numbers of coalitions, KernelSHAP and IME do a poor job of describing model behavior based on ablation tests. Even with more coalitions, we find that they still underperform in the NHANES loss explanation (Figure 25b and 25c) and MNIST feature extraction (Figure 26b) examples. Although we would expect the unbiased stochastic estimators to perform strongly with more coalitions, each of the three experiments takes roughly 12 hours to run. Increasing the number of coalitions may lead to experiments that take days, if not weeks, to compute.

Although these stochastic estimators may work reasonably well in simple settings with small numbers of features as in the HELOC model stacking example (Figure 27b), they are still dramatically slower and it can be quite difficult to assess convergence. Furthermore, the HELOC setting constitutes a distributed model setting where model agnostic estimators may be undesirable because they require sharing proprietary models.

These results suggest that, in our datasets and models, the bias incurred using DeepSHAP can have a smaller impact on ablation tests compared to the variance in model agnostic estimators. We incorporate the above discussion and the following figures in a new *Appendix Section A.5.9 “Evaluating convergence of unbiased estimators”*. We hope that our more extensive empirical comparisons to model agnostic approaches can convince the reviewer that DeepSHAP’s fast, but biased explanations can be a valuable and complementary tool to the slower unbiased model agnostic estimators, particularly for explaining a series of models.

Figure 25: Convergence of unbiased stochastic estimators on the NHANES loss explanation example. We explain 10 explicands using 1000 baselines in terms of their loss attributions based on the same model from Figures 5c and 5d. We compute the DeepSHAP (deep) attributions once since they are deterministic. Then, we compute the IME (samp) and KernelSHAP (kern) attributions ten times for each number of coalitions to capture their variability. (a) We visualize the average standard deviation per feature across the ten replicates of stochastic estimates. (b) The negative loss ablations. (c) The positive loss ablations. (b) and (c) In the opaque colors we visualize the mean across the ten replicates. In the transparent colors we visualize each of the ten replicates.

Figure 27: Convergence of unbiased stochastic estimators on the HELOC model stack example. We explain 100 explicands using 100 baselines in terms of their output attributions based on the same model from Figure 7. We compute the DeepSHAP (deep) attributions once since they are deterministic. Then, we compute the IME (samp) and KernelSHAP (kern) attributions ten times for each number of coalitions to capture their variability. (a) We visualize the average standard deviation per feature across the ten replicates of stochastic estimates. (b) Ablating the top 10% negative and positive features. On the left, we visualize ablating the top 10% negative features, for which higher Δ s are better. On the right, we visualize ablating the top 10% positive features, for which lower Δ s are better. We show the 95% confidence intervals based on the variability of the ablations across replicates.

Figure 28: Runtimes of approaches from convergence experiments. We report the runtimes of computing explanations based on Figures 25, 26, and 27. We also show 95% confidence intervals for IME (samp) and KernelSHAP (kern). For DeepSHAP (deep), we only run it once and visualize it’s runtime as a horizontal line because it does not depend on the number of coalitions parameter.

They build on top of the interventional Shapley values, which break the correlation structure in the data (i.e. the data manifold). And, I do not think they sufficiently justify why explaining a series of models in their proposed way would actually be superior in practical contexts (e.g. consumers of the explanations being non-technical, the features used in upstream models being unknown or difficult to interpret).

Thank you for the excellent recommendation. In order to highlight these practical benefits we have incorporated a new paragraph in Section 1 “Introduction” Lines 66-73:

In practice, DeepSHAP can use feature attributions to ask many important scientific questions by explaining different parts of the series of models (Figure 1d). When features used by upstream models are semantically meaningless (deep feature extraction) or hard to understand (stacked generalization), DeepSHAP provides explanations in terms of the original features which can often be more intuitive, especially for non-technical consumers. In addition, DeepSHAP enables attributions with respect to different aspects of model behavior such as predicted risk or even errors the model makes (loss explanation). Finally, using the group rescale rule enables users to reduce the dimensionality of highly correlated features which makes them easier to understand (group explanation).

As a result, I would not recommend the publication of this work in Nature Communications.

Though it does not address my main concern, and thus could not change my recommendation, I would like to commend the authors for their comprehensive empirical study, including helpful qualitative commentary alongside data sets requiring more expertise to interpret (e.g. genetics).

We thank the reviewer for the valuable feedback which has helped us to improve our manuscript. Once again, we hope that our more extensive empirical comparisons to model agnostic approaches can

convince the reviewer that DeepSHAP's fast, but biased explanations can be a valuable and complementary tool to the slower unbiased model agnostic estimators, particularly for explaining a series of models.

Reviewer #4 (Remarks to the Author):

Overall, I think the authors have written a solid manuscript. I would be willing to recommend the paper for publication after the authors have made the revisions outlined in the review that follows.

In this paper the authors present DeepSHAP, a method for explaining the predictions of complex multi-layered machine learning models using a model-agnostic local feature attribution method based on Shapley values. The Shapley values method has become popular within the explainable AI community, as it benefits from a number of game-theoretical optimality guarantees. An initial version of the proposed methodology has been introduced in the authors' previous paper as Deep SHAP (reference [17]), a combination between DeepLIFT and their SHAP method for computing Shapley values. The approach presented here is an improvement over the original method incorporating refinements and new results from the literature.

We thank the reviewer for their very thorough and helpful feedback on our analyses which have given us the opportunity to improve our manuscript. The reviewer's suggestions and concerns are addressed below.

The methodology appears sound and is largely based on combining previously published results from the literature on explainable AI. The main advantage of DeepSHAP seems to lie in its flexibility. The method can provide local attributions in a number of different ways (e.g., group attributions, intermediary attributions, model loss, different baselines), it can work on mixed ML model types, and it can accommodate series of models, in which each model belongs to a separate institution. In my opinion, the authors make some good arguments and provide highly relevant examples for showing why this flexibility is necessary. Another advantage of the proposed method lies in its very quick computation time. While the showcases results are impressive, I think that the authors should stress a bit more that the reason why the method is so fast is because it only provides approximate interventional Shapley values.

We thank the reviewer for the great suggestion. We add an explicit description of DeepSHAP as an approximation in *Section 2 "Generalizing DeepSHAP local explanations" Lines 99-104.*

DeepSHAP is an approximate method, meaning that it is biased for the true interventional Shapley values. However, this bias allows DeepSHAP to be drastically faster than alternative approaches. This strategy of trading bias for speed is taken by other Shapley value estimators including L-Shapley [31], C-Shapley [31], Deep Approximate Shapley Propagation [32], and Shapley Explanation Networks [33]. To ensure the attributions are valuable despite this bias, we extensively evaluate DeepSHAP both qualitatively and quantitatively in the following sections.

31. Chen, J., Song, L., Wainwright, M. J. & Jordan, M. I. L-shapley and c-shapley: Efficient model interpretation for structured data. arXiv preprint arXiv:1808.02610 (2018).

32. Ancona, M., Oztireli, C. & Gross, M. Explaining deep neural networks with a polynomial time algorithm for shapley value approximation in International Conference on Machine Learning (2019), 272–281.
33. Wang, R., Wang, X. & Inouye, D. I. Shapley Explanation Networks. arXiv preprint arXiv:2104.02297 (2021).

Furthermore, we note that trading off bias in exchange for speed is interestingly a common theme in a few Shapley value estimators.

In particular, L-Shapley and C-Shapley impose a constraint based on neighborhoods nearby features [33]. This imposition requires spatial correlation assumptions, and enables polynomial runtime in C-Shapley and exponential runtime in the neighborhood size in L-Shapley. We do not include comparisons to L-Shapley and C-Shapley because although they are model agnostic, spatial correlation assumptions only make sense for data such as images and text data. Instead, DeepSHAP is meant to be more flexible and accommodate tabular data as in Figures 3, 4, 5, and 7.

Two additional methods that rely on assumptions to produce relatively fast, but biased estimates of Shapley values are Deep Approximate Shapley Propagation (DASP) [34] and Shapley Explanation Networks (ShapNets) [35]. DASP produces estimates for baseline shapley in $O(d^2)$ model evaluations, where d is the number of features. This is relatively slow in comparison to DeepLIFT which requires a single backward pass (on the order of a single model evaluation). Ultimately DASP’s estimates are still biased, because the uncertainty propagation they rely on requires assumptions that are not quite true; however, they show lower bias relative to interventional Shapley values in comparison to DeepLIFT. ShapNets produce estimates for baseline Shapley in a single model evaluation and have no bias for Shallow ShapNets, a variant that only includes a single hidden layer and has bias for Deep ShapNets, a variant that includes multiple hidden layers.

We do not include comparisons to DASP and ShapNets for three reasons. (1) They require very specific model architectures. DASP requires first and second order moment matching for each layer, which is not known in general. ShapNets require utilization of their specific architecture where each hidden node can only have a very small number of inputs (2 or 3 typically) and Shapley estimates are built into the model. (2) DASP and ShapNets are designed to produce baseline Shapley value estimates and cannot generate interventional Shapley value estimates as in DeepSHAP. (3) Our experiments primarily encompass model stacks that include more than just deep models, in which case DASP and ShapNets are not applicable.

We include the discussion above in a new *Appendix Section A.5.2 “Examples of Shapley value estimators that trade bias for speed” Lines 995-1018*:

L-Shapley and C-Shapley impose a constraint based on neighborhoods nearby features [33]. This constraint depends on spatial correlation assumptions, and enables polynomial runtime in the neighborhood size for C-Shapley and exponential runtime in the neighborhood size for L-Shapley. We do not include comparisons to L-Shapley and C-Shapley because although they are model agnostic, spatial correlation assumptions only make sense for data such as images and text data. Instead, DeepSHAP is meant to be more flexible and accommodate tabular data as in Figures 3, 4, 5, and 7.

Two additional methods that rely on assumptions to produce relatively fast, but biased estimates of Shapley values are Deep Approximate Shapley Propagation (DASP) [34] and Shapley Explanation Networks (ShapNets) [35]. DASP produces estimates for baseline shapley in $O(N^2)$ model evaluations, where N is the number of features. This is relatively slow in comparison to DeepLIFT which requires a single backward pass (on the order of a single model evaluation). Ultimately DASP's estimates are still biased, because the uncertainty propagation they rely on requires assumptions that are not quite true; however, they show lower bias relative to interventional Shapley values in comparison to DeepLIFT. ShapNets produce estimates for baseline shapley in a single model evaluation and have no bias for Shallow ShapNets, a variant that only includes a single hidden layer and has bias for Deep ShapNets, a variant that includes multiple hidden layers.

We do not include comparisons to DASP and ShapNets, for three reasons. (1) They require very specific model architectures. DASP requires first and second order moment matching for each layer, which is not known in general. ShapNets require utilization of their specific architecture where each hidden node can only have a very small number of inputs (2 or 3 typically) and Shapley estimates are built into the model. (2) DASP and ShapNets are designed to produce baseline Shapley value estimates and cannot generate interventional Shapley value estimates as in DeepSHAP. (3) Our experiments primarily encompass model stacks that include more than just deep models, in which case these techniques cannot be applied since they are specific to deep models.

33. Chen, J., Song, L., Wainwright, M. J. & Jordan, M. I. L-shapley and c-shapley: Efficient model interpretation for structured data. arXiv preprint arXiv:1808.02610 (2018).
34. Ancona, M., Oztireli, C. & Gross, M. Explaining deep neural networks with a polynomial time algorithm for shapley value approximation in International Conference on Machine Learning (2019), 272–281.
35. Wang, R., Wang, X. & Inouye, D. I. Shapley Explanation Networks. arXiv preprint arXiv:2104.02297 (2021).

I think that the authors should explicitly write down the improvements made to DeepSHAP since it was initially proposed.

We thank the reviewer for highlighting this point of confusion.

In the original SHAP paper, they aim to calculate observational Shapley values. However, due to the difficulty in estimating observational conditional expectations, they actually calculate interventional Shapley values (although the connection to interventional conditional expectations had yet to be pointed out) [32, 55].

DeepSHAP was originally introduced as an adaptation of DeepLIFT in the original SHAP paper [17] designed to make DeepLIFT closer to the interventional Shapley values. However, it is briefly and informally introduced, making it difficult to know exactly what the method entails and how it differs from DeepLIFT. The old DeepSHAP is the same as DeepLIFT, but with the reference (baseline) values set to the average of the baseline samples. In comparison, DeepLIFT typically sets the baseline to uninformative values, and sets them to zeros for image data (an all-black image).

However, interventional Shapley values are equivalent to an average of baseline Shapley values, but not to baseline Shapley values with the average as a baseline. Due to this interpretation (**Section 6.2**), it is more natural to calculate DeepSHAP as the average of many attributions for different baselines. This in turn allows us to formulate a generalized rescale rule which allows us to propagate attributions through pipelines of linear, tree, and deep models for which baseline Shapley values are easy to calculate.

In order to clarify the differences, we explicitly define DeepSHAP as it was originally briefly proposed in [17] and the current version we are proposing:

- DeepSHAP (old): rescale rule with average baseline.
- DeepSHAP (current): generalized rescale rule and group rescale rule with multiple baselines.

In terms of applications, the old DeepSHAP only applies to deep models. The current version applies to pipelines of linear, tree, and deep models. Finally, the group rescale rule gives us a natural approach to group large numbers of features and thus generate attributions for a much smaller number of groups. This type of sparsity is often helpful for helping humans understand model explanations [63].

To clarify these differences, we include references to this section in multiple sections (“*Generalizing DeepSHAP local explanations*” and “*Interventional Shapley values baseline distribution*” Lines 84 and 395) to a new *Methods Section 6.8 “Differences to previous approaches”* Lines 477-499:

In the original SHAP paper [17], they aim to calculate observational Shapley values. However, due to the difficulty in estimating observational conditional expectations, they actually calculate what is later described as interventional Shapley values [32, 55].

DeepSHAP was originally introduced as an adaptation of DeepLIFT in the original SHAP paper [17] designed to make DeepLIFT closer to the interventional Shapley values. However, it is briefly and informally introduced, making it difficult to know exactly what the method entails and how it differs from DeepLIFT. The old DeepSHAP is the same as DeepLIFT, but with the reference (baseline) values set to the average of the baseline samples. In comparison, DeepLIFT typically sets the baseline to uninformative values, and sets them to zeros for image data (an all-black image).

However, interventional Shapley values are equivalent to an average of baseline Shapley values, but not to baseline Shapley values with the average as a baseline. Due to this interpretation (Section 6.2), it is more natural to calculate DeepSHAP as the average of many attributions for different baselines. This in turn allows us to formulate a generalized rescale rule which allows us to propagate attributions through pipelines of linear, tree, and deep models for which baseline Shapley values are easy to calculate.

In order to clarify the differences, we explicitly define DeepSHAP as it was originally briefly proposed in [17] and the current version we are proposing. The old DeepSHAP used the rescale

rule with an average baseline. The current DeepSHAP uses the generalized rescale rule and group rescale rule with multiple baselines. In terms of applications, the old DeepSHAP only applies to deep models. The current version applies to pipelines of linear, tree, and deep models. Finally, the group rescale rule gives us a natural approach to group large numbers of features and thus generate attributions for a much smaller number of groups. This type of sparsity is often helpful for helping humans understand model explanations [63].

17. Lundberg, S. M. & Lee, S.-I. A unified approach to interpreting model predictions in Advances in Neural Information Processing Systems (2017), 4765–4774.

32. Janzing, D., Minorics, L. & Blöbaum, P. Feature relevance quantification in explainable AI: A causality problem. arXiv preprint arXiv:1910.13413 (2019).

55. Chen, H., Janizek, J. D., Lundberg, S. & Lee, S.-I. True to the Model or True to the Data? arXiv preprint arXiv:2006.16234 (2020).

63. Lipton, Z. C. The Mythos of Model Interpretability: In machine learning, the concept of interpretability is both important and slippery. Queue 16, 31–57 (2018).

On lines 297-298, the authors mention an evaluation of their method by Schwab & Karlen, but it is not clear to which version of the method are they referring.

Thank you for addressing this point of confusion. Schwab & Karlen use DeepSHAP (rescale rule) with multiple baselines. This can be viewed as an application of the current version of DeepSHAP specifically to a deep model. This is because the first publicly available implementation of DeepSHAP actually averages over attributions from many baselines, despite being originally introduced as using a single average baseline [17]. We clarify this point in *Section 5 “Discussion” Lines 315-316*:

17. Lundberg, S. M. & Lee, S.-I. A unified approach to interpreting model predictions in Advances in Neural Information Processing Systems (2017), 4765–4774.

Schwab and Karlen show that for explaining deep networks, **our version of DeepSHAP, which uses multiple baselines**, is a very fast yet performant approach in terms of an ablation test for explaining MNIST and CIFAR images.

38. Schwab, P. & Karlen, W. CXPlain: Causal explanations for model interpretation under uncertainty in Advances in Neural Information Processing Systems(2019), 10220–10230.

One difference is that compared to the original Deep SHAP method, the authors are now using the interventional Shapley values proposed by Janzing (reference [30]) instead of (conditional) Shapley values. Additionally, the authors developed a generalized DeepLIFT rescale rule with the purpose of extending the method's application to models other than neural networks. They also propose a group rescale rule that allows the method to aggregate the attributions to higher level groups of features.

Thank you, this is a more explicit description of the improvements since the presentation of the old DeepSHAP algorithm in the original SHAP paper [17]. We have incorporated these, as mentioned above in a new *Methods Section 6.8 “Differences to previous approaches”*. One caveat is that the original SHAP paper described the conditional (observational) Shapley values, but in fact, they actually estimate

interventional Shapley values. This includes the original DeepSHAP method which was never really an estimator for conditional (observational) Shapley values.

17. Lundberg, S. M. & Lee, S.-I. A unified approach to interpreting model predictions in Advances in Neural Information Processing Systems (2017), 4765–4774.

I also think that the authors could make a stronger case for why they have settled for interventional Shapley values. The authors mention that this approach for computing Shapley values "is most closely related to DeepSHAP", and it seems to me that the crucial benefit of the interventional approach is their ability to decompose into a number of baseline Shapley values.

We thank the reviewer for raising an excellent point.

The differences between the interventional Shapley values and the observational Shapley values is a common point of debate. In fact, this point is the topic of numerous papers [14, 22, 36, 37, 53]. Some advocate for the interventional Shapley values and others advocate for the observational Shapley values. Although the observational Shapley values are what is advocated for in the original SHAP paper, they actually estimate interventional Shapley values. The primary reason is that the observational Shapley values require an estimate of the observational conditional expectation which is very hard to obtain in general.

In our paper, we aim to estimate the interventional Shapley values because they do not spread credit among correlated features and do a better job identifying features the models algebraically depend on [52]. In contrast, observational Shapley values will spread credit among correlated features [53] and give a view of the information content each feature has with regard to the output.

Although spreading credit can be desirable, it can lead to counterintuitive attributions. For example, we can consider the scenario where we have two identical features x_1 and x_2 and we train a model that only uses feature x_1 . Interventional Shapley values will tell us that the model does not rely on feature x_2 and it will have zero attribution. In contrast, observational Shapley values will tell us that x_1 and x_2 both matter equally because they are highly correlated even though the model does not use x_2 to generate predictions [14].

Although it can still be arguable which of these approaches is preferable, it should be noted that this issue of spreading importance between correlated features is not unique to Shapley values but is rather a fundamental issue encountered in every removal-based feature attribution method [54].

For our paper, we prefer the interventional approach as it is closer to understanding the model's behavior, which can be useful for understanding good and bad models. When used to understand good models (models that the user believes to have learned meaningful patterns), it can be a tool to discover interesting non-linear associations between features and the output. When used to understand bad models it can be a tool to debug models and see where they fail. In contrast, the observational approach aims to understand the information content each feature has with regards to the output, which can be hard to understand for models with poor performance. Furthermore, if the user really wants to spread credit between correlated features, it is possible to do so during the model fitting stage (rather than the explanation stage). Using

regularization or ensemble methods, interventional Shapley values will naturally spread credit as observational Shapley values do [53].

A second motivation to use interventional Shapley values is that estimating the observational conditional expectation is drastically harder than the interventional conditional expectation. This is reflected in a wide disagreement about how to estimate the observational conditional expectation [14, 37, 53, 54, 55, 56, 57, 58, 59]. On the other hand, the interventional conditional expectation has one agreed upon empirical estimation strategy [14, 21]. This difficulty also reflects in the model-specific approaches, where there are exact algorithms to calculate interventional Shapley values for linear and tree models (LinearSHAP [53] and TreeSHAP [21]). In particular, TreeSHAP and DeepSHAP are both based on the useful property that interventional Shapley values decompose into an average of baseline Shapley values (Section 6.2). As the reviewer points out, this benefit is crucial to the design of the generalized rescale rule.

We have incorporated this discussion in *Section 6.1 “The Shapley value” Lines 356-375*:

Both approaches have tradeoffs that have been described elsewhere [14, 22, 36, 37, 53]. Here, we focus on the interventional approach for two primary reasons:

- 1. Observational Shapley values will spread credit among correlated features 1531. Although this can be desirable, it can lead to counterintuitive attributions. In particular, features that the model literally does not use to calculate its predictions will have non-zero attribution simply if they are correlated with features the model heavily depends on 1141. Instead, the interventional Shapley values do a better job of identifying the features the models algebraically depend on 1531. As such, the interventional Shapley values are useful for debugging bad models and drawing insights from good models. In contrast, observational Shapley values are hard to understand for models with poor performance. Finally, if it is really important to spread credit using correlated features, it is possible to modify the model fitting using regularization or ensembles which will cause interventional Shapley values to naturally spread credit 1531.**
- 2. Estimating the observational conditional expectation is drastically harder than the interventional conditional expectation. This is reflected in a wide disagreement about how to estimate the observational conditional expectation 1541, with approaches including empirical 1141, cohort refinement 114, 55, 561, parametric assumptions 153, 561, generative model 1371, surrogate model 1371, missingness during training 1541, separate models 157–591. On the other hand, the interventional conditional expectation has one agreed upon empirical estimation strategy 114, 211. This difficulty also reflects in the model-specific approaches, where there are exact algorithms to calculate interventional Shapley values for linear and tree models (LinearSHAP 1531 and TreeSHAP 1211). In particular, TreeSHAP and DeepSHAP are both based on the useful property that interventional Shapley values decompose into an average of baseline Shapley values (Section 6.2). This benefit is crucial to the design of the generalized rescale rule.**

14. Sundararajan, M. & Najmi, A. The many Shapley values for model explanation in Proceedings of the International Conference on Machine Learning (2020), 513–523.
 21. Lundberg, S. M. et al. Explainable AI for Trees: From Local Explanations to Global Understanding. CoRR abs/1905.04610. arXiv: 1905.04610. <http://arxiv.org/abs/1905.04610> (2018).
 22. Merrick, L. & Taly, A. The Explanation Game: Explaining Machine Learning Models Using Shapley Values in International Cross-Domain Conference for Machine Learning and Knowledge Extraction (2020), 17–38.
 36. Kumar, I. E., Venkatasubramanian, S., Scheidegger, C. & Friedler, S. Problems with Shapley-value-based explanations as feature importance measures. arXiv preprint arXiv:2002.11097 (2020).
 37. Frye, C., de Mijolla, D., Cowton, L., Stanley, M. & Feige, I. Shapley-based explainability on the data manifold. arXiv preprint arXiv:2006.01272 (2020).
 53. Chen, H., Janizek, J. D., Lundberg, S. & Lee, S.-I. True to the Model or True to the Data? arXiv preprint arXiv:2006.16234 (2020).
 54. Covert, I., Lundberg, S. & Lee, S.-I. Explaining by removing: A unified framework for model explanation. Journal of Machine Learning Research 22, 1–90 (2021).
 55. Mase, M., Owen, A. B. & Seiler, B. Explaining black box decisions by Shapley cohort refinement. arXiv preprint arXiv:1911.00467 (2019).
 56. Aas, K., Jullum, M. & Loland, A. Explaining individual predictions when features are dependent: More accurate approximations to Shapley values. arXiv preprint arXiv:1903.10464 (2019).
 57. Lipovetsky, S. & Conklin, M. Analysis of regression in game theory approach. Applied Stochastic Models in Business and Industry 17, 319–330 (2001).
 58. trumbelj, E., Kononenko, I. & ikonja, M. R. Explaining instance classifications with interactions of subsets of feature values. Data & Knowledge Engineering 68, 886–904 (2009).
 59. Williamson, B. & Feng, J. Efficient nonparametric statistical inference on population feature importance using Shapley values in International Conference on Machine Learning (2020), 10282–10291.
-

I would have liked to see a comparison to the explanations produced by the original Deep SHAP method, proposed in [17]. What are the consequences of choosing this new approach when it comes to the explanations provided? From a theoretical perspective, (conditional) Shapley values and interventional Shapley values offer rather different ways of looking at the feature attribution.

We thank the reviewer for bringing up this point. To briefly reiterate, the original version of DeepSHAP is not actually tied to the observational (conditional) Shapley values. Instead, it may be better viewed as an approximation to the baseline Shapley values with an average baseline. This is because under linearity assumptions, the old DeepSHAP formulation exactly matches the baseline Shapley values, but not the observational Shapley values.

In comparison, the current version of DeepSHAP first generates attributions for many baselines and then averages them which serves as an approximation to the interventional Shapley values. Again this is because under linearity assumptions, the current formulation of DeepSHAP exactly matches the interventional Shapley values. To better clarify these differences we incorporate a new *Section 6.8 “Differences to previous approaches”*.

In addition, we demonstrate that an average baseline (Old DeepSHAP) leads to biased attributions similarly to a zero baseline (DeepLIFT). In *Appendix Figure 10*, we show the average baseline used to explain each image: for CIFAR it is a blurred image but it will change depending on the dataset. In the “Delta of Image and Average Baseline” column, the dark pixels represent parts of the image that are very close to the baseline. We find that using the average baseline strongly biases the attribution to not place importance on these areas of the explicand that resemble the average baseline. More generally, any single

baseline can easily encounter explicands where pixels that match the baseline will be biased to have very low attribution.

An additional practical detriment to using an average baseline is that it simply does not work when using categorical features, because it is unclear what the average is.

We include a reference in *Section 6.8* to a new *Appendix Section A.4.2* “*Bias for an average baseline*”
Lines 928-933:

We compare DeepSHAP to the original formulation presented in [17]. The original formulation was equivalent to DeepLIFT with a single average baseline. However, this approach and any approach which depends on a single baseline is susceptible to bias. We demonstrate this in Figure 10, where using an average baseline biases the attribution to give low importance to parts of the image which resemble the baseline (dark areas in “Delta of Image and Average Baseline”).

Figure 10: Using a single average baseline image (Old DeepSHAP) leads to biased attributions. The image is the explicand. The average baseline showcases average across 1000 randomly sampled baselines. The “Delta of Image and Average Baseline” shows the absolute difference between the image and the average baseline summed across channels. Images in this column will have darker pixels in regions where the explicand closely matches the average baseline. The attribution plots are the sum across channels of the absolute value of the feature attributions.

If the intention is to also give a causal interpretation to the explanations, then it might be worthwhile to have a look at the discussion in the paper on “Causal Shapley Values” (Heskes et al., 2020).

We thank the reviewer for the opportunity to highlight this connection. We are primarily interested in using the interventional Shapley values because it is both tractable and useful for understanding model

behavior, not because it has a causal interpretation. We include a citation to [60] and describe the differences to the interventional Shapley values in *Methods Section “6.1 The Shapley value” Lines 376-381*:

Note that a third approach, named causal Shapley values, uses causal inference's interventional conditional expectation, but does not assume a flat causal graph [60]. Causal Shapley values require knowledge of a causal graph relating the input variables and the output. However, in general this graph is unknown or requires substantial domain expertise. In addition, causal Shapley values can be hard to estimate and require estimating many interventional probabilities. In contrast, interventional Shapley values are a tractable way to understand model behavior.

60. Heskes, T., Sijben, E., Bucur, I. G. & Claassen, T. Causal shapley values: Exploiting causal knowledge to explain individual predictions of complex models. arXiv preprint arXiv:2011.01625(2020).

Perhaps the strongest aspect of the manuscript is the extensive experimental evaluation of the method. The authors have shown that the method can provide valuable insights into a number of datasets coming from unrelated domains (genetics, finance, computer vision). They compare DeepSHAP against a number of relevant competitors (IME, KernelSHAP, LIME) and show how their method can achieve a good explanatory performance through ablation experiments in a short amount of time. The practicality of the method suggests to me that this work has the potential to be a significant addition to the field.

We thank the reviewer for the positive feedback.

The paper is generally well written and structured. I would like to commend the authors for their nice illustrations, which, in my opinion, greatly facilitate the understanding of the paper. However, I would like to draw their attention to the very end of the paper, right before the Acknowledgments, where it seems that they have not finished the last paragraph on ablation tests. Please make sure to fix that issue in the final version.

We thank the reviewer for catching this incomplete paragraph. We have completed the sentence in question as follows:

In contrast, for negative ablations, as we ablate the most negative features, better attributions will cause the mean model output to increase rapidly and lead to higher curves.

Regarding reproducibility, another important issue is that no code has been provided. The link mentioned by the authors in Section 10 does not seem to work, at least at the time of the reviewing. Apart from that, the authors seem to have provided sufficient detail in the main paper and in the appendix to allow for the work to be reproduced.

We thank the reviewer for pointing this out. We had forgotten to make the repository public and have done so now.

Other comments:

- Page 2, after Equation (1): When explaining the composition of functions, it is unclear to me what the "o_i" functions are and how they are connected to the "h_i" functions.

We thank you for bringing up this point. We have corrected a typo in the methods section (changing $h_{i=o_{i-1}}$ to $m_{i=o_{i-1}}$) and added a sentence clarifying that m_i are the input dimensions and o_i are the output dimensions for each layer (or model) h_i .

- Page 2, footnote: I am not familiar with the term "flat causal graph", and it is not an established term to the best of my knowledge. Could the authors please provide the definition in the manuscript revision?

Thank you for bringing up this point of clarification. We have added a definition of a flat causal graph as a causal graph where arrows are only drawn between input variables and the output in footnote 1 in Section 1 "Introduction" and Section 2 "Generalizing DeepSHAP local explanations" Lines 86-87.

- Page 6, line 158: "neurodegenerative" -> "neurodegenerative"

- Page 7, line 181: "Figure 3a" -> "Figure 5a"

- Page 8, line 183: "Figure 3b" -> "Figure 5b"

- Page 11, Figure 7: "Explanation of bank model" should be subfigure (c) and "Explanation of full pipeline" should be subfigure (d).

Fixed - thank you.

- Page 17, Equation (35): The authors should also explain the notation for the Hadamard product. Using x^b for both the rows and for the matrix might be confusing so maybe use X^b instead?

Thank you, we have added these to the revision.

- Page 18, line 451: The code for the experiments does not seem to be available at the link mentioned by the authors.

We thank the reviewer for pointing this out. We had forgotten to make the repository public and have done so now.

REVIEWERS' COMMENTS

Reviewer #2 (Remarks to the Author):

The authors thoroughly answered the questions. For scientists familiarized with feature attribution methods/SHAP, the manuscript is educative, analyses are complete, code is provided so that methods are reproducible, and DeepSHAP has the potential to be useful in different fields. A negative aspect is the general accessibility of the manuscript. Even though applications are presented for data sets from different domains, differences between the previous and the newly reported DeepSHAP approach are probably not accessible to a general/multidisciplinary audience.

Reviewer #4 (Remarks to the Author):

Thank you to the authors for their detailed response to the issues the other reviewers and I have raised. In my original review, I expressed my belief that the manuscript is a solid piece of research, and after the rebuttal I still think the article is worthy of publication. That being said, my co-reviewers have raised some excellent points, which I think the authors have done their best to address.

I appreciate that the authors have now included paragraphs describing the improvements of the current approach relative to the 'Deep SHAP' method from the original paper. Perhaps using a different name could also further help to distinguish the two approaches. The name DeepSHAP also suggests that this method is only suitable for explaining the predictions of deep learning models, whereas in fact it can apply to more general model pipelines. I also appreciate the fact that the authors have now added the necessary clarification that their method is biased and constitutes a fast approximation of interventional Shapley values. It is unfortunate that "no theoretical arguments about the bias" can be made, but the results seem to indicate that the bias-variance trade-off is beneficial. The manuscript is improved by the inclusion of references to other approximate methods (L-Shapley, C-Shapley, DASP, ShapNets).

I would also like to thank the authors for clarifying that in the original SHAP paper, interventional Shapley values are actually estimated, since it is an important point of confusion. I think the authors have made a decent case for using interventional over observational Shapley values, highlighting how difficult they are to estimate and how they spread credit among correlated features. However, I would like to point out that the claim "In contrast, observational Shapley values are hard to understand for models with poor performance." is not sufficiently explained. I understand that the reason for this is the fact that observational Shapley values "give a view of the information content each feature has with regard to the output", which is hard to do when the model is poor. I would add this reason to the text,

otherwise the above claim seems empty of content. Perhaps adding a reference at that point would also solidify the claim.

I agree with Reviewer #2 that "an interesting aspect of the manuscript is how the baseline distribution can influence the local feature attributions" and I appreciate that the authors have taken the time to explain the fine differences between the various Shapley value estimation methods (e.g., averaged versus baseline, observational versus interventional) in more detail. I agree with Reviewer #3 that it would be preferable to have a method that is "highly justified and standardised", but at the same time I believe that heuristical approaches have their merit and can later lead to a better understanding of why a particular approach performs well in a certain task. I think the authors have at least showcased through extensive experimental evaluation that the output of their method can be useful.

Reviewer #2 (Remarks to the Author):

The authors thoroughly answered the questions. For scientists familiarized with feature attribution methods/SHAP, the manuscript is educative, analyses are complete, code is provided so that methods are reproducible, and DeepSHAP has the potential to be useful in different fields.

We thank the reviewer for their positive feedback.

A negative aspect is the general accessibility of the manuscript. Even though applications are presented for data sets from different domains, differences between the previous and the newly reported DeepSHAP approach are probably not accessible to a general/multidisciplinary audience.

In terms of the accessibility of the manuscript, we make the following modifications to include a high-level description of the differences between our approach and previous approaches in Section 1.1 “Generalizing DeepSHAP local explanations”.

In this paper, we improve upon two previous approaches [17, 18] that propagate attributions while maintaining efficiency with respect to a single baseline. We make two improvements: (1) we compare to a distribution of baselines, which decreases the reliance of the attributions on any single baseline (Section 3.1) and (2) we generalize the rescale rule so that it applies to series of mixed model types, rather than only layers in a deep model.

Reviewer #4 (Remarks to the Author):

Thank you to the authors for their detailed response to the issues the other reviewers and I have raised. In my original review, I expressed my belief that the manuscript is a solid piece of research, and after the rebuttal I still think the article is worthy of publication. That being said, my co-reviewers have raised some excellent points, which I think the authors have done their best to address.

We thank the reviewer for their positive feedback.

I appreciate that the authors have now included paragraphs describing the improvements of the current approach relative to the 'Deep SHAP' method from the original paper. Perhaps using a different name could also further help to distinguish the two approaches. The name DeepSHAP also suggests that this method is only suitable for explaining the predictions of deep learning models, whereas in fact it can apply to more general model pipelines.

We thank the reviewer for the excellent suggestion. We do agree that changing the name could be helpful for distinguishing the two approaches. In addition to distinguishing the approaches in Methods Section 4.8 we have also changed the name of our approach to Generalized DeepSHAP, (G-DeepSHAP). We have modified this through the manuscript and figures.

I also appreciate the fact that the authors have now added the necessary clarification that their method is biased and constitutes a fast approximation of interventional Shapley values. It is unfortunate that "no theoretical arguments about the bias" can be made, but the results seem to indicate that the bias-variance trade-off is beneficial. The manuscript is improved by the inclusion of references to other approximate methods (L-Shapley, C-Shapley, DASP, ShapNets).

I would also like to thank the authors for clarifying that in the original SHAP paper, interventional Shapley values are actually estimated, since it is an important point of confusion. I think the authors have made a decent case for using interventional over observational Shapley values, highlighting how difficult they are to estimate and how they spread credit among correlated features. However, I would like to point out that the claim "In contrast, observational Shapley values are hard to understand for models with poor performance." is not sufficiently explained. I understand that the reason for this is the fact that observational Shapley values "give a view of the information content each feature has with regard to the output", which is hard to do when the model is poor. I would add this reason to the text, otherwise the above claim seems empty of content. Perhaps adding a reference at that point would also solidify the claim.

We thank the reviewer for pointing out this lack of justification. We add the following justification in Section 4.1 "The Shapley value".

As such, the interventional Shapley values are useful for debugging bad models and drawing insights from good models. In contrast, **although observational Shapley values give a view of the information content each feature has with regard to the output for optimal models [56], this is not the case for bad models. Furthermore, it can be hard to use observational Shapley values to debug bad models because it is unclear whether a feature is important because it is explicitly depended on by the model or because it is correlated with the features the model explicitly depends on.**

56. Covert, I., Lundberg, S. & Lee, S.-I. Explaining by removing: A unified framework for model explanation. Journal of Machine Learning Research 22, 1–90 (2021).

I agree with Reviewer #2 that "an interesting aspect of the manuscript is how the baseline distribution can influence the local feature attributions" and I appreciate that the authors have taken the time to explain the fine differences between the various Shapley value estimation methods (e.g., averaged versus baseline, observational versus interventional) in more detail. I agree with Reviewer #3 that it would be preferable to have a method that is "highly justified and standardised", but at the same time I believe that heuristical approaches have their merit and can later lead to a better understanding of why a particular approach performs well in a certain task. I think the authors have at least showcased through extensive experimental evaluation that the output of their method can be useful.